# Disentangling Interpretable Factors with Supervised Independent Subspace Principal Component Analysis

**Jiayu Su**[1,2,5†]    **David A. Knowles**[2,4,5†]    **Raul Rabadan**[1,2,3†]

[1]Program for Mathematical Genomics; [2]Department of Systems Biology, Columbia University
[3]Department of Biomedical Informatics, Columbia University
[4]Department of Computer Science, Columbia University    [5]New York Genome Center
† Correspondence to {js5756, rr2579}@cumc.columbia.edu, dak2173@columbia.edu

## Abstract

The success of machine learning models relies heavily on effectively representing high-dimensional data. However, ensuring data representations capture human-understandable concepts remains difficult, often requiring the incorporation of prior knowledge and decomposition of data into multiple subspaces. Traditional linear methods fall short in modeling more than one space, while more expressive deep learning approaches lack interpretability. Here, we introduce *Supervised Independent Subspace Principal Component Analysis (sisPCA)*, a PCA extension designed for multi-subspace learning. Leveraging the Hilbert-Schmidt Independence Criterion (HSIC), sisPCA incorporates supervision and simultaneously ensures subspace disentanglement. We demonstrate sisPCA's connections with autoencoders and regularized linear regression and showcase its ability to identify and separate hidden data structures through extensive applications, including breast cancer diagnosis from image features, learning aging-associated DNA methylation changes, and single-cell analysis of malaria infection. Our results reveal distinct functional pathways associated with malaria colonization, underscoring the essentiality of explainable representation in high-dimensional data analysis.

## 1 Introduction

High-dimensional data generated by complex biological mechanisms encapsulate an ensemble of patterns. A prime example is single-cell RNA sequencing (scRNA-seq) data (Fig. 1). These datasets measure the expression of tens of thousands of genes across potentially millions of cells, creating rich tapestries woven from interacting cellular pathways, gene dynamics, cell states, and inherent measurement noise. To unravel these patterns and reveal hidden relationships, it is necessary to decompose the data into meaningful, lower-dimensional subspaces.

Linear representation learning methods, such as Principal Component Analysis (PCA) [Hotelling, 1933] and Independent Component Analysis (ICA) [Comon, 1994], extract latent spaces from data using explainable linear transformations. These widely employed unsupervised tools learn a single latent space or a union of one-dimensional subspaces. Independent Subspace Analysis (ISA) extends ICA by extracting multidimensional components as independent subspaces [Cardoso, 1998]. Yet, the unsupervised nature of these methods precludes knowledge integration, restricting their utility and sometimes even identifiability [Theis, 2006]. In the context of Fig. 1, the representation learned without supervision fails to separate temporal variability from technical batch effects.

Conversely, recent advancements in deep generative models, especially semi-supervised approaches [Kingma et al., 2014], have shown promise in disentangling diverse latent spaces and retaining relevant information under supervision. However, challenges remain in ensuring subspace independence within a variational autoencoder (VAE). While models like $\beta$-VAE [Higgins et al., 2016], HCV

38th Conference on Neural Information Processing Systems (NeurIPS 2024).

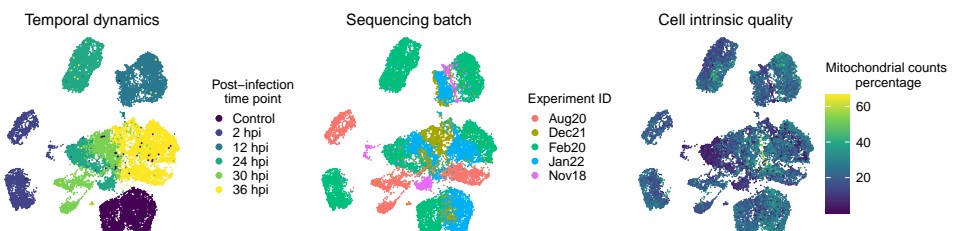

Figure 1: Example scRNA-seq dataset from Afriat et al. [2022]. Each dot represents the gene expression vector $\vec{x} \in \mathbb{R}^{8,203}$ of a cell, visualized in 2D and colored by cell properties $\{Y_m\}$. Variability in the dataset $X$ arises from multiple sources: (left to right) temporal dynamics of infection, technical batch effects, and cell quality. Incorporating supervisory information $Y$, such as time points, allows for the extraction of patterns in distinct subspaces $\{Z_m\}$ that correspond to different sources of variability. Moreover, the linear mapping $\{U_m : X \to Z_m\}$ directly quantifies the relationship between gene expression and the property of interest, enabling discoveries such as the identification of genes underlying the persistent defense against infection. The disentanglement is particularly important to ensure minimal confounding effects. See Section 4.4 for details.

[Lopez et al., 2018] and biolord [Piran et al., 2024] attempt to address this, inference for deep generative models remains challenging and the learned representations are not interpretable.

To bridge the gap, we propose Supervised Independent Subspace Principal Component Analysis (`sisPCA`)[1], an innovative method extending PCA to *multiple subspaces*. By incorporating the Hilbert-Schmidt Independence Criterion (HSIC), `sisPCA` effectively decomposes data into explainable independent subspaces that align with target supervision. It thus reconciles the simplicity and clarity of linear methods with the nuanced multi-space modeling capabilities of advanced generative models. In summary, our contributions include:

- A multi-subspace extension of PCA for disentangling linear latent subspaces in high-dimensional data. We additionally show that supervising subspaces with a linear target kernel can be conceptualized as linear regression regularized akin to Zellner's g-prior.

- An efficient eigendecomposition-based alternating optimization algorithm to compute `sisPCA` subspaces. The learning process with linear kernels resembles matrix factorization, which may potentially benefit from desirable local geometric properties (Conjecture 3.1).

- Demonstrated effectiveness and interpretability of `sisPCA` in various applications. This includes identifying diagnostic image features for breast cancer, dissecting aging signatures in human DNA methylation data, and unraveling time-independent transcriptomic changes in mouse liver cells upon malaria infection.

## 2 Background

### 2.1 Hilbert-Schmidt Independence Criterion (HSIC)

The Hilbert-Schmidt Independence Criterion (HSIC) serves as a methodology for testing the independence of two random variables $X$ and $Y$ [Gretton et al., 2005a]. It operates by embedding probability distributions into reproducing kernel Hilbert spaces (RKHS) and quantifying independence by distance between the joint distribution and the product of its marginals. Specifically, HSIC builds on the cross-covariance operator $C_{XY} : \mathcal{G} \to \mathcal{F}$, which is a generalization of the cross-covariance matrix to infinite-dimensional RKHS (i.e., the feature spaces of $X$ and $Y$ after transformation),

$$C_{XY} := \mathbb{E}_{XY}[\phi(X) \otimes \psi(Y)] - \mathbb{E}_X[\phi(X)] \otimes \mathbb{E}_Y[\psi(Y)].$$

Here $\mathcal{F} := \text{span}(\{\phi(X)\})$ and $\mathcal{G} := \text{span}(\{\psi(Y)\})$ are the RKHSs with feature maps $\phi$ and $\psi$ for $X$ and $Y$ respectively. The tensor product operator $f \otimes g$ maps $\mathcal{G}$ to $\mathcal{F}$, such that $(f \otimes g)h := f\langle g, h \rangle_{\mathcal{G}}$ for all $h \in \mathcal{G}$. The HSIC is then defined as,

$$HSIC(X, Y; \mathcal{F}, \mathcal{G}) := ||C_{XY}||^2_{HS} := \sum_{i,j} \langle C_{xy} v_i, u_j \rangle_{\mathcal{F}}$$

---

[1]A Python implementation of `sisPCA` is available on GitHub at https://github.com/JiayuSuPKU/sispca (DOI 10.5281/zenodo.13932660). The repository also includes notebooks to reproduce results in this paper.

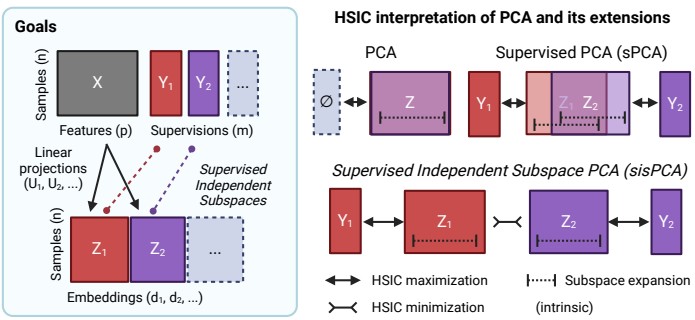

Figure 2: Overview of `sisPCA` and its relationship with other PCA models.

where $v_i$ and $u_j$ are the orthogonal bases of $\mathcal{G}$ and $\mathcal{F}$. $X$ and $Y$ are independent if and only if $HSIC(X, Y; \mathcal{F}, \mathcal{G}) = 0$. In practice, given a finite sample $\{(x_i, y_i)\}_{i=1}^{n}$ from the joint distribution $P_{X,Y}$, the empirical HSIC can be computed as,

$$HSIC_n(X, Y) = \frac{1}{(n-1)^2} tr(K_X H L_Y H)$$

where $K_X$ and $L_Y$ are matrices of kernel evaluations over the samples $\{x_i\}_{i=1}^{n}$ in $\mathcal{F}$ and $\{y_i\}_{i=1}^{n}$ in $\mathcal{G}$, and $H$ is the centering matrix defined as $H = I_n - \frac{1}{n} \mathbf{1}_n \mathbf{1}_n^\top$, with $I_n$ being the identity matrix.

## 2.2 Related work

The HSIC, as a non-parametric criterion for independence, has been effectively integrated into numerous representation learning models, especially for disentangling complex data structures. One of the earliest applications of HSIC is ICA [Comon, 1994], where the goal is to recover unmixed and statistically independent sources. Gretton et al. [2005a] showed that minimizing HSIC via gradient descent outperforms specialized linear ICA algorithms. Alternatively, HSIC can also be maximized to encode specific information in learning tasks. In supervised PCA (sPCA), HSIC is deployed to guide the identification of principal subspaces with maximum dependence on target variables [Barshan et al., 2011]. More recently, Ma et al. [2020] used HSIC as a supervised learning objective for deep neural networks to bypass back-propagation, and Li et al. [2021] subsequently extend it to the context of self-supervised learning.

Our primary interest lies in extending these models to identify *multiple subspaces*, each representing independent, meaningful signatures. In this direction, Cao et al. [2015] suggested the inclusion of HSIC as a diversity term in multi-view subspace clustering, encouraging representations from different views to capture complementary information. Building on a similar idea, Lopez et al. [2018] incorporated HSIC-based regularization into VAE architectures to promote subspace separation. However, there has yet to be a linear multi-space model for more interpretable data decomposition.

The presented work is also related to contrastive representation learning, as introduced in Abid et al. [2018] and Abid and Zou [2019]. Along this line of research, recent studies have also applied HSIC to regularize contrastive VAE subspaces [Tu et al., 2024, Qiu et al., 2023]. See Appendix A for more discussions on connections and differences.

## 3 The sisPCA model

We introduce *Supervised Independent Subspace Principal Component Analysis (sisPCA)*, a linear model for disentangling independent data variation (Fig. 2). The model's linearity ensures explicit interpretability and enables regression-based extensions such as sparse feature selection. We formally discuss the connection between `sisPCA` and regularized linear regression in Section 3.2.

### 3.1 Problem formulation

As motivated in Fig. 1, given a dataset $\{X_i \in \mathcal{X}^p\}_{i=1}^{n}$ with $n$ observations and $p$ features and $m$ associated target variables $\{Y_i \in \mathcal{Y}^m\}_{i=1}^{n}$, we aim to find $m$ separate subspace representations of

the data $\{\{\mathcal{Z}_i^j \subset \mathbb{R}^{d_j}\}_{j=1}^m\}_{i=1}^n$ with latent dimensions $\{d_j\}_{j=1}^m$. Each subspace should maximize dependence with one target variable while minimizing dependence with other subspaces. For simplicity, we assume Euclidean space, $\{X_i\} := X \in \mathbb{R}^{n \times p}$ and $\{Y_i\} := Y \in \mathbb{R}^{n \times m}$. Other types of data can be easily handled using appropriate kernels, which will be discussed later. The linear projection to the $j$-th subspace $\mathcal{U}_j : \mathbb{R}^p \to \mathbb{R}^{d_j}$ is represented by the matrix $U_j \in \mathbb{R}^{p \times d_j}$, with $Z_j := XU_j \in \mathbb{R}^{n \times d_j}$ being the data representation in this $j$-th subspace. Similar to the concept of PCA loading, the projection $U_j$ depicts linear combinations of original features in $X$ and thus can be directly interpreted as feature importance scores.

Our overall objective is to find the set of subspace projections $\{U_1, ..., U_m\}$ that solves

$$\operatorname*{argmax}_{U_1,...,U_m} \sum_{j=1}^m \mathcal{I}(XU_j, Y_j) - \lambda \sum_{j=1}^m \sum_{i>j}^m \mathcal{I}(XU_i, XU_j),$$

under some constraints. Here $\mathcal{I}$ is a measure of dependence and $\lambda$ penalizes overlapping subspaces. Using HSIC as the dependence measure, `sisPCA` solves the constrained optimization

$$\operatorname*{argmax}_{U_1,...,U_m} \sum_{j=1}^m tr(K_{Z_j} H K_{Y_j} H) - \lambda \sum_{j=1}^m \sum_{i>j}^m tr(K_{Z_i} H K_{Z_j} H) \tag{1}$$

$$\text{subject to } U_j^T U_j = I, \ \forall j \in \{1, ..., m\},$$

where $K_{Z_j}$ and $K_{Y_j}$ are kernels defined on the $j$-th subspace $\mathcal{Z}_j$ and the $j$-th target variable $\mathcal{Y}_j$, respect. This formulation differs from kernel PCA, where in the first term of (1) the kernel is defined over $Z$ rather than $X$ (the kernel extension of `sisPCA` will be discussed later).

In the special case where $K_{Z_j} := Z_j Z_j^T$ is linear for all subspaces, the optimization becomes

$$\operatorname*{argmax}_{U_1,...,U_m} \sum_{j=1}^m tr(XU_j U_j^T X^T H K_{Y_j} H) - \lambda \sum_{j=1}^m \sum_{i>j}^m tr(XU_i U_i^T X^T H XU_j U_j^T X^T H) \tag{2}$$

$$\text{subject to } U_j^T U_j = I, \ \forall j \in \{1, ..., m\}.$$

The first term is the supervised PCA objective [Barshan et al., 2011]. We can thus view the above formulation, termed `sisPCA-linear`, as an extension of supervised PCA to multiple subspaces with additional regularization for subspace independence (Fig. 2). It comes with an appealing property:

*Remark* 3.1. *Maximizing the `sisPCA-linear` objective* (2) *is equivalent to minimizing the reconstruction error of a linear autoencoder plus regularization. See Appendix B.*

We now examine the second HSIC term in (2). Consider two subspaces $Z_u := XU \in \mathbb{R}^{n \times d_u}$ and $Z_v := XV \in \mathbb{R}^{n \times d_v}$ with centered $X$. The HSIC regularization is

$$tr(XUU^T X^T XVV^T X^T) = tr(Z_u Z_u^T Z_v Z_v^T) = ||Z_u^T Z_v||_F^2 \geq 0.$$

This term equals zero if and only if $Z_u$ and $Z_v$ are orthogonal. While not convex in $U$ and $V$ jointly, the coupling of $U$ and $V$ solely through the matrix product $Z_u^T Z_v$ indicates a well-behaved local geometry, as explored by Sun and Luo [2016]. Indeed, it allows `sisPCA-linear` to benefit from theoretical insights on matrix factorization. For example, Ge et al. [2017] showed that for a quadratic function $f$ over the matrix $U^T V$ — in our context $f = ||Z_u^T Z_v||_F^2$ — all local minima are also globally optimal under mild conditions achievable through proper regularization.

Returning to (2), we see that spurious local optima may only emerge from subspace imbalance in the first symmetry-breaking supervision term, where some subspaces may contribute more to the overall objective. This leads to the following conjecture on the optimization landscape of (2):

**Conjecture 3.1** (informal): *The `sisPCA-linear` objective* (2) *has no spurious local optima under balanced supervision, which is achievable through proper regularization; With unbalanced supervision, the global optima can still be recovered using local search algorithms following simple initialization based on the relative supervision strength of each subspace.*

Intuitively, the balanced supervision condition is to ensure that the local optima induced by subspace symmetry and interchangeability are also global optima. For more discussions on the optimization landscape, the balance condition, and Conjecture 3.1, see Appendix C.

We now consider an iterative optimization approach to solve (2).

*Remark* 3.2. *The optimization problem in* (2) *can be solved via alternating optimization, where each iteration has an analytical update for each subspace.*

Using basic matrix algebra, the objective of (2) simplifies to,

$$\sum_{j=1}^{m} tr(U_j^T X^T H K_{Y_j} H X U_j) - \frac{\lambda}{2} \sum_{j=1}^{m} \sum_{i \neq j}^{m} tr(K_{Z_i} H K_{Z_j} H) = \sum_{j=1}^{m} tr(U_j^T X^T \tilde{K}_j X U_j)$$

where $\tilde{K}_j := H(K_{Y_j} - \frac{\lambda}{2} \sum_{i=1, i \neq j}^{m} K_{Z_i})H$ and $K_{Z_i} := Z_i Z_i^T = X U_i U_i^T X^T$. Given the set $\{U_{i \neq j}\}_i$, $U_j$ can be updated by maximizing $tr(U_j^T X^T \tilde{K}_j X U_j)$, leading to the update $U_j^{(t+1)} \leftarrow Q_{d_j}$, where $Q_{d_j}$ are the columns of $Q$ corresponding to the $d_j$ largest eigenvalues from the eigendecomposition $X^T \tilde{K}_j X := Q \Lambda Q^T$.

The full optimization process is outlined in Algorithm 1, Appendix B. This procedure guarantees convergence to an optimum as the objective is bounded and non-decreasing in every iteration. We further implement an initialization step to find the path (subspace update order) towards the global optimum as proposed in Conjecture 3.1. Briefly, we compare subspace contributions to the supervision loss and prioritize updates for subspaces under stronger supervision.

While convenient, a zero HSIC regularization loss with a linear kernel in (2) does not guarantee independent subspaces. For strict independence, the subspace kernel $K_Z$ in (1) needs to be universal (e.g., Gaussian). We refer to this as `sisPCA-general` and solve it using gradient descent (Algorithm 2, Appendix D). The naive implementation with a Gaussian kernel has complexity $O(n^3)$ in contrast to $O(n^2)$ with a linear kernel. See Appendix D for performance difference discussions.

Both `sisPCA-linear` and `sisPCA-general` are linear methods where $\{U_m\}$ measures direct contribution of original features to each subspace. Nevertheless, the `sisPCA` framework can be easily extended to incorporate nonlinear feature interactions, analogous to kernel PCA. See Appendix E.

## 3.2 Kernel selection for different target variables

The use of kernel independence measure in (1) allows `sisPCA` to accommodate various data types through flexible kernel choices for $K_X$ (data), $K_Z$ (latent subspace), and $K_Y$ (target).

For categorical variables $Y = Y_j{}_{i=1}^{n}$ (e.g., cancer types), we use the Dirac delta kernel,

$$K_Y(i,j) = \mathbb{1}_{Y_i = Y_j}.$$

It is also possible to use other general graph kernels for categorical variables with an intrinsic hierarchical structure, e.g., subtypes and stages [Smola and Kondor, 2003].

For continuous variables $Y \in \mathbb{R}^n$, we use the linear kernel

$$K_Y = YY^T.$$

When $Y \in \mathbb{R}^{n \times d}$ is multivariate, the kernel is the sum of per-dimension kernels $K_Y = \sum_{i=1}^{d} Y_{:i} Y_{:i}^T$.

*Remark* 3.3. *Maximizing the `sisPCA-linear` objective* (2) *with linear kernels on the target space is equivalent to performing regularized regression against the target. See Appendix B.*

In a nutshell, we show that `sisPCA-linear` can be viewed as approximating the target $Y$ with $Z = Xu$. The particular regularization on $u$ corresponds to a zero-mean multivariate Gaussian prior, which is related to Zellner's g-prior in Bayesian regression.

## 3.3 Learning an unknown subspace without supervision

In practical applications, a common goal would be to recover both subspaces linked to known attributes (supervised) and to unknown attributes (unsupervised) simultaneously. In `sisPCA`, this is achieved by setting the target kernel for the unknown subspace to the identity matrix ($K_Y = I$), corresponding to unsupervised PCA (Fig. 2). Section 4.1 provides an example of this process.

However, the absence of external supervision introduces a potential identifiability issue. As indicated in Appendix B eq. 4 each supervised subspace is driven by two forces to (1) align with the target and (2) capture major variations in the data. This dual objective can lead to scenarios where unknown attributes, ideally retained in the residual unsupervised subspace, being inadvertently presented in supervised subspaces. Fig. 8 in Appendix D gives an example where `sisPCA-general` fails.

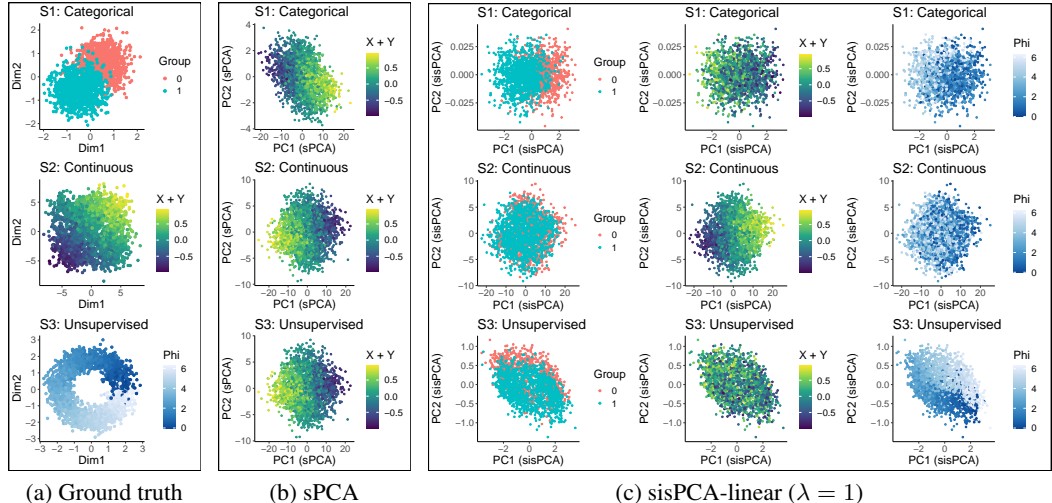

Figure 3: Example application of recovering a latent space with three subspaces (rows in panel a) embedded in a high-dimensional space. The first two subspaces (rows) of sPCA (panel b) and `sisPCA` (panel c) are supervised by the corresponding target variables.

## 4 Applications

Unless otherwise specified, in the following sections `sisPCA` refers to `sisPCA-linear`, where the objective (2) is solved using Algorithm 1. For baseline comparisons, we consider linear models including PCA and sPCA. While non-linear VAE counterparts such as HCV [Lopez et al., 2018] are included for quantitative performance benchmarking in Section 4.4, they are not considered for interpretability analyses due to their inherent complexity. The rationale and details for baseline selection are provided in Appendix F.

### 4.1 Recovering supervised and unsupervised subspaces in simulated data

We first consider learning latent subspaces associated with known and unknown attributes using simulated data. The dataset reflects a ground truth 6-dimensional latent space, comprising three distinct 2D subspaces (Fig. 3a): S1 with two Gaussian distributions, S2 with a noisy 2D grid and S3 with a ring structure. The defining manifold characteristics $\phi$ of S3 remain unknown to the model, representing the unsupervised component. These subspaces were concatenated and linearly projected to 20-dimensions using a $6 \times 20$ matrix with entries uniformly distributed on $[0, 1]$.

Both unsupervised PCA (targeting S3) and supervised PCA (S1 and S2) subspaces capture structures heavily influenced by S2 (Fig. 3b and Fig. 11 in Appendix H), due to S2's pronounced variations. In contrast, `sisPCA` markedly improves the disentanglement of these subspaces, especially the unsupervised S3 (Fig. 3c). Despite S2's dominant influence, `sisPCA` isolates the effects of each subspace, resulting in clearer separation of the three independent signals. The two supervised subspaces S1 and S2 are distinctly characterized by patterns exclusively associated with the supervision attributes. In the unsupervised subspace S3, although Principal Component 2 (PC2) picks up some categorical information from S1, `sisPCA` successfully uncovers the underlying circular structure.

### 4.2 Learning diagnostic subspaces from breast cancer image features

We apply `sisPCA` to the Kaggle Breast Cancer Wisconsin Data[2] to demonstrate its utility in data compression and feature extraction. The dataset contains 569 samples with 30 summary features from breast mass imaging. Our goals are to (1) learn compressed subspaces for predicting disease status ('Malignant' or 'Benign', agnostic during training), and (2) understand the relationship between original features and how they contribute to the learned representation and diagnosis potential.

---

[2]uciml/breast-cancer-wisconsin-data, CC BY-NC-SA 4.0 license.

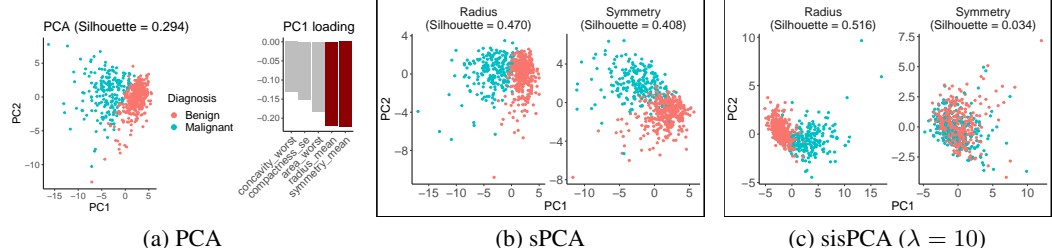

Figure 4: Feature extraction on the breast cancer dataset. The two top PC1 contributors in PCA (panel a) are used as supervisions to construct the 'radius' and 'symmetry' subspaces (panel b and c).

The diagnosis label, invisible to all models, is used to measure subspace quality via the mean silhouette score $\frac{1}{n} \sum_i s(i) = \frac{1}{n} \sum_i (b(i) - a(i)) / \max\{a(i), b(i)\}$, where $a(i)$ and $b(i)$ are mean intra-cluster and nearest-cluster distances for sample $i$. Higher scores indicate larger diagnostic potential. In addition, we use the Geodesic Grassmann distance $d(Z_i, Z_j) = \left( \sum_{n=1}^{k} \theta_n^2 \right)^{1/2}$ to measure subspace separateness, where $k = \min\{\dim Z_i, \dim Z_j\}$ and $\{\theta_n\}$ the principal angles [Miao and Ben-Israel, 1992]. Higher scores indicate better disentanglement. Inputs for all model are zero-centered and variance-standardized.

In the PCA space, samples are well-separated based on diagnosis along PC1 (Fig. 4a). 'symmetry_mean' and 'radius_mean' are the top two features negatively contributing to PC1, motivating us to construct separate subspaces to reflect nuclei size (using 'radius_mean' and 'radius_sd' as targets) and shape (using 'symmetry_mean' and 'symmetry_sd' as targets). The remaining 26 features are projected onto these subspaces using sPCA (Fig. 4b) and `sisPCA` (Fig. 4c). In sPCA, both subspaces better explain diagnosis status than PCA but remain highly entangled (Grassmann distance: $1.493$, Pearson correlation of PC2 loading: $0.850$). However, with `sisPCA`'s explicit disentanglement, the symmetry subspace loses its predictive power as the two spaces separate further (Grassmann distance: $2.710$). `sisPCA` subspaces are constructed from distinct feature sets (PC2 loading correlation $-0.203$), with 'area' and 'perimeter' contributing more to the radius subspace and 'compactness' and 'smoothness' to the symmetry one. The radius subspace also gains additional predictive power by repulsing further from the symmetry space (Silhouette scores: $0.516$ in `sisPCA`, $0.470$ in sPCA).

Our `sisPCA` results suggest that cell nuclear size is more informative for breast cancer diagnosis than nuclear shape. We confirm this by measuring directly the predictive potential of target variables (Silhouette scores: 0.457 for 'radius_mean' and 'radius_sd', 0.092 for 'symmetry_mean' and 'symmetry_sd'). Our conclusion also aligns with previous clinical observations [Kashyap et al., 2018]. In contrast, PCA and sPCA, while capable to extract new diagnostic features (PC1), cannot faithfully capture feature relationships without disentanglement and potentially overestimate symmetry-related features' relevance in malignancy.

### 4.3 Separating aging-dependent DNA methylation changes from tumorigenic signatures

Tumorigenesis and aging are two intricately linked biological processes, resulting in cancer omics data that often display patterns of both. DNA methylation (DNAm) exemplifies this complexity, undergoing genome-wide alterations during aging while also exhibiting cancer-specific changes in particular regions, presumably silencing tumor suppressor genes or activating oncogenes.

#### 4.3.1 Problem and dataset description

The Cancer Genome Atlas (TCGA)[3] offers a comprehensive collection of DNAm datasets from patients with various cancer types. In TCGA DNAm data, methylation status is probed across genomic locations using the Illumina Infinium 450K array and quantified as beta values ranging from 0 (unmethylated) to 1 (methylated). The resulting data matrix $X$ presents challenges due to the use of three different probe types and highly correlated features, typically requiring careful preprocessing. For illustration purpose, we use a downsampled dataset comprising the first 5,000 non-constant and non-NA CpG sites from 9,725 TCGA tumor samples across 33 cancer types.

---

[3]Data access is controlled through the NCI Genomic Data Commons (https://portal.gdc.cancer.gov/).

Our goal is to disentangle tumorigenic signatures from age-dependent methylation dynamics in this pan-cancer DNAm data. Traditional methylation analyses often employ regression-based methods to learn site-specific statistics, later aggregated by gene or high-level genomic annotations [Bock, 2012]. However, these approaches may suffer from high dimensionality and multi-collinearity due to potential redundancy in CpG methylation activity. We propose using `sisPCA` to address these limitations, which allows us to (1) learn compressed, low-dimensional representations that retain biological information, and (2) minimize confounding factors not controlled in simple regression models by enforcing disentanglement.

Specifically, we aim to learn two subspaces: one aligning with chronological age (CA, aging subspace, supervised with a linear kernel) and another with TCGA cancer categories (cancer subspace, supervised with a delta kernel). We evaluate the quality of learned representations using information density measured by the Silhouette score and subspace separateness by the Grassmann distance. For the rank-one aging subspace $(\mathrm{rank}(K_Y) = \mathrm{rank}(YY^T) = 1)$, we also measure information density using the maximum absolute Spearman correlation, $\max_{d\in[1,d_j]}\{|\rho(Z_j^{(d)}, \mathrm{CA})|\}$, between CA and each axis of the subspace $Z_j \in \mathbb{R}^{n \times d_j}$.

### 4.3.2 Quantitative performance of subspace quality

We first validate the disentanglement effect of formulation (2). Unlike models such as Lopez et al. [2018] that use a Gaussian kernel, `sisPCA` minimizes the HSIC regularization with a linear kernel. This approach trades strict statistical guarantees for improved computational efficiency (detailed in Appendix D). Our experiments demonstrate that as $\lambda$ increases, the two subspaces show increasing divergence (Table 1). Notably, while HSIC-Gaussian is not explicitly optimized, it decreases in tandem with HSIC-Linear. The generally larger values of HSIC-linear potentially offer advantages in optimization and help mitigate numerical rounding errors.

Table 1: Separateness of the aging and cancer subspaces inferred by `sisPCA`.

| | | sisPCA | | |
| --- | --- | --- | --- | --- |
| | PCA | $\lambda = 0$ (sPCA) | $\lambda = 1$ | $\lambda = 10$ |
| HSIC-Linear (in the objective of `sisPCA`) | 481.6 | 189.7 | 2.0e-4 | **2.4e-05** |
| HSIC-Gaussian | 1.1e-2 | 6.5e-3 | 7.0e-4 | **7.0e-4** |
| Grassmann distance | 0 | 3.09 | 4.97 | **4.97** |

Furthermore, the separation of aging and cancer subspaces leads to a moderate increase in target information density and a decrease in confounding information (Table 2). However, stronger regularization does not always equate to better representations. This is partly due to the inherent coupling between aging and tumorigenesis. Efforts to remove aging signals inevitably result in information loss on cancer type. We discuss the tuning of $\lambda$ more generally in Appendix G.

Table 2: Information density in each `sisPCA` subspace.

| | | | sisPCA | | |
| --- | --- | --- | --- | --- | --- |
| | Subspace | PCA | $\lambda = 0$ | $\lambda = 1$ | $\lambda = 10$ |
| Maximum Spearman correlation with age | age | 0.213 | 0.278 | 0.286 | **0.294** |
| | cancer | 0.213 | 0.233 | **0.103** | 0.115 |
| Silhouette score with cancer type | age | 0.074 | -0.183 | -0.221 | **-0.230** |
| | cancer | 0.074 | 0.106 | **0.107** | 0.097 |

### 4.4 Disentangling infection-induced changes in the mouse single-cell atlas of the *Plasmodium* liver stage

Malaria, transmitted by mosquitoes carrying the *Plasmodium* parasite, involves a critical liver stage where the parasite colonizes and replicates within host hepatocytes. This section examines the intricate host-parasite interactions at single-cell resolution during this stage.

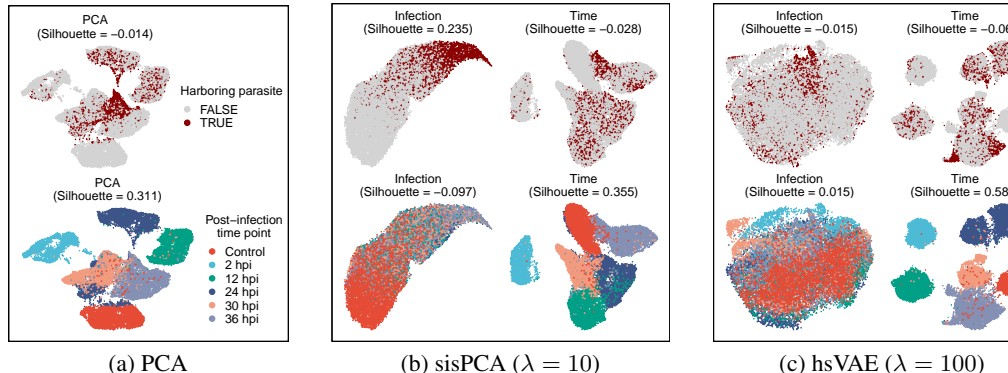

Figure 5: UMAP visualizations of scRNA-seq data. Each column shows a different learned subspace: (a) PCA, (b) `sisPCA`-infection and `sisPCA`-time, and (c) hsVAE-infection and hsVAE-time. See Fig. 12 for other models. Cells are colored by either infection status (top row) or post-infection time (bottom row). In an optimal pair of subspaces, each property (infection status or time) should be more distinguishable in its corresponding subspace while showing less separation in the other.

### 4.4.1 Problem and dataset description

We analyze scRNA-seq data of mouse hepatocytes from Afriat et al. [2022][4]. Our goal is to distinguish genes directly involved in parasite harboring from those associated with broader temporal changes post-infection. The processed dataset comprises gene expression profiles of 19,053 cells collected at five post-infection time points (2, 12, 24, 30, and 36h) and from control mice. Infection status was determined based on GFP expression linked to malaria. We keep the top 2,000 highly variable genes and use normalized expression as model inputs. Time points are treated as discrete categories to account for potential nonlinear dynamics. See Appendix F for full experiment details.

### 4.4.2 Learning the infection subspace associated with parasite encapsulation

UMAP visualizations of PCA, sPCA, and `sisPCA` subspaces show that while PCA primarily captures temporal variations (Fig. 5a), both sPCA and `sisPCA` successfully differentiate between infected and uninfected hepatocytes in their infection subspaces. However, sPCA's infection space still exhibits significant temporal effects, suggesting uncontrolled confounding effects (Fig. 12b in Appendix H). `sisPCA` effectively eliminates this intermingling, yielding cleaner representations where relevant biological information is further enriched in the corresponding spaces (Fig. 5b).

Comparisons with non-linear VAE counterparts (Fig. 5c and Fig. 12, full model description in Appendix F) and quantitative evaluations (Table 3) demonstrate that our HSIC-based supervision formulation (1) achieves performance comparable to neural network predictors. Notably, `sisPCA` outperforms HSIC-constrained supervised VAE (hsVAE)[Lopez et al., 2018] in separating infected and uninfected cells. Indeed, hsVAE's performance in the infection subspace is so poor that even under supervision, the representation contains near-random information (Fig. 5c). To address this gap, we developed hsVAE-sc by incorporating additional domain-specific knowledge (See Appendix F, and Fig. 12d in Appendix H). This model learns a much improved infection space (Silhouette score: 0.233) while maintaining high distinguishability in the temporal space (Silhouette score: 0.634). The improvement is likely due to the fact that parasite encapsulation can induce a slight increase in total RNA counts; by using unnormalized count-level data and explicitly modeling the library size, the model captures this additional information and thus produces enhanced results.

While visualizations and quantitative metrics provide useful estimates, they may not fully capture the biological relevance of the representations. For instance, a subspace with two point masses perfectly separating cells by infection status would have the highest information density, yet offer little new biological insight. Advantageously, linear models like `sisPCA` are inherently interpretable. Using the learned `sisPCA` projection $U$ as feature importance scores, we rank genes based on their PC contributions. Chemokine genes, including *Cxcl10*, emerge as top positive contributors to infection-PC1, which is elevated in infected cells. This aligns with their established role as acute-

---

[4]Preprocessed data available at https://doi.org/10.6084/m9.figshare.22148900.v1 (CC BY 4.0).

Table 3: Quantitative evaluation of subspace representation quality.

| | | Linear | | | Non-linear | | |
|---|---|---|---|---|---|---|---|
| | Subspace | PCA | sPCA | **sisPCA** | VAE | supVAE | hsVAE |
| Grassmann distance | | 0 | 3.771 | **4.824** | 0 | 3.571 | 3.598 |
| Silhouette - infection | infection | -0.014 | 0.207 | **0.235** | -0.036 | 0.041 | -0.015 |
| | time | -0.014 | -0.075 | **-0.097** | -0.036 | -0.087 | -0.069 |
| Silhouette - time point | infection | 0.311 | **-0.029** | **-0.028** | 0.296 | 0.052 | 0.015 |
| | time | 0.311 | 0.348 | 0.355 | 0.296 | 0.479 | **0.582** |

phase response markers. Gene Ontology (GO) enrichment analysis of genes with significant PC1 loading scores reveals that infection leads to reduced fatty acid metabolism and enhanced stress and defense responses, consistent with known *Plasmodium* harboring effects. These results remain highly consistent across a wide range of hyperparameters, demonstrating the robustness of our approach. See Appendix G for full examination on the effect of $\lambda$.

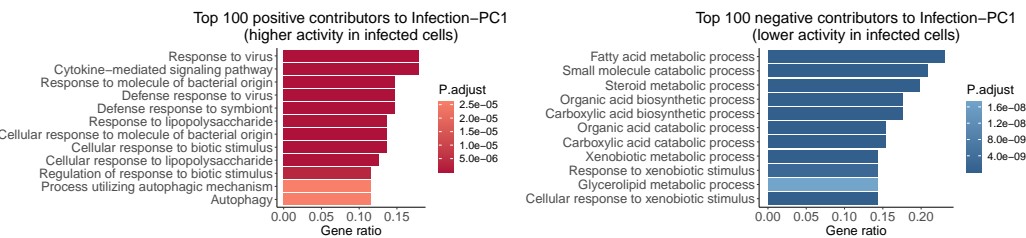

Figure 6: GO biological process enrichment results of top genes contributing to the `sisPCA`-infection subspace. Genes are ranked by their PC1 loading and are grouped by effect direction.

## 5 Discussion

This study presents `sisPCA`, a novel extension of PCA for disentangling multiple subspaces. We showcase its capability in interpretable analyses for learning complex biological patterns, such as aging dynamics in methylation and transcriptomic changes during *Plasmodium* infection. To enhance usability, we have implemented an automatic hyperparameter tuning pipeline for $\lambda$ using grid search and spectral clustering, similar to contrastive PCA [Abid et al., 2018] (Appendix G).

Still, `sisPCA` has several limitations: **Linearity constraints:** The linear nature of `sisPCA` may miss non-linear feature interactions, potentially underperforming on more complicated datasets. While nonlinear extensions are possible (Appendix E), they come at the cost of reduced computational efficiency and interpretability. **Linear kernel HSIC:** The HSIC-linear regularization, while computationally convenient, does not guarantee complete subspace independence. However, minimizing HSIC-linear tends to reduce HSIC with a Gaussian kernel (Table 1), which suggests that the issue is less of a concern in practice. **Subspace identifiability:** Our formulation (1) relies on external supervision to differentiate subspaces, which could lead to identifiability issues if the supervisions are too similar, or when one subspace is unsupervised.

Despite these limitations, `sisPCA`'s ability to provide interpretable, disentangled representations of complex biological data makes it a valuable addition to the toolkit of biologists working with high-dimensional datasets. We envision future applications on larger-scale omics datasets and potential new biomedical discoveries.

## 6 Acknowledgements

We thank Bianca Dumitrascu for early discussions and anonymous reviewers for their valuable comments. This work was funded by the National Institutes of Health, National Cancer Institute (R35CA253126, U01CA261822, and U01CA243073 to R. R.) and the Edward P. Evans Center for MDS at Columbia University (to J. S. and R. R.).

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

# A  Connections with contrastive representation learning

Contrastive representation learning, as introduced in Abid et al. [2018], aims to find representations that capture the difference between explicit positive and negative samples. For example, contrastive PCA (cPCA) takes the pair of a target dataset of interest and a background dataset as inputs and finds the principal components that account for large variances in the target but small variances in the background [Abid et al., 2018]. cVAE [Abid and Zou, 2019] and contrastiveVI [Weinberger et al., 2023] apply the idea to VAE architectures, where the target and background data are assumed to have different generative processes and the corresponding representations lie in separate subspaces. The learning of the target and background subspaces are achieved by explicitly forcing target and background samples to use different spaces. More recently, Tu et al. [2024] and Qiu et al. [2023] adopt an HSIC-based regularization approach to encourage the disentanglement of the two subspaces.

Despite the similarity, our proposed `sisPCA` model is conceptually different from these contrastive models. `sisPCA` (and the non-linear HCV [Lopez et al., 2018]) focuses on the decomposition of a *single* dataset, as opposed to a *pair* of target and background datasets. The sense of contrast in `sisPCA` comes from *explicit supervision*, of which each subspace either aligns with or is repulsed from. If we concatenate the pair of target-background into one dataset and consider the case-control information as supervision, the associated `sisPCA` subspace can also reflect the difference similar to cPCA (the objective is still not the same).

# B  Properties of the sisPCA-linear optimization problem

## B.1  Proof of Remark 3.1 on the equivalence of `sisPCA-linear` and autoencoder

Here we show that the `sisPCA-linear` optimization 2 is equivalent to solving a regularized linear autoencoder. First note the well-known fact of PCA that maximizing the variance of the projection $||XU||_F^2$ is equivalent to minimizing the reconstruction error of a linear orthogonal autoencoder, since

$$||X - XUU^T||_F^2 = tr(X^T X - 2X^T XUU^T + X^T UU^T UU^T X)$$
$$= tr(X^T X) - tr(U^T X^T XU) = ||X||_F^2 - ||XU||_F^2.$$

Now consider supervised PCA with a positive semidefinite kernel $K_y$ and its Cholesky decomposition $K_y := L^T L$. Note $L := I$ in vanilla unsupervised PCA. That is, PCA can be viewed as aligning to a supervision kernel where every sample forms a group of itself. Without loss of generality, we assume $X$ and $K_y$ to be centered, such that the first term in eq.2 per subspace becomes

$$tr(XUU^T X^T K_y) = tr(U^T X^T K_y XU) = ||LXU||_F^2.$$

Maximizing the above quantity is thus equivalent to minimizing the weighted reconstruction error $||L(X - XUU^T)||_F^2$ where $L$ from supervision controls the aspects to focus on.

## B.2  The alternating optimization algorithm for sisPCA-linear

Below we describe the Algorithm 1 to solve (2).

## B.3  Proof of Remark 3.3 on the equivalence of sisPCA-linear and regression

Here we show that the `sisPCA-linear` problem 2 is equivalent to a regularized linear regression. For simplicity, we assume both $X$ and $Y$ to be centered and $Y \in \mathbb{R}^{n \times 1}$ to be univariate. Since $X^T K_Y X = X^T YY^T X$ is rank-one, we need only to consider the largest eigenvalue and corresponding eigenvector of $X^T K_Y X$. Therefore, maximizing the first cross-covariance term $tr(U^T X^T K_Y XU) = ||Y^T XU||_F^2$ in (2) is equivalent to finding a unit vector $u \in \mathbb{R}^{p \times 1}$ by

$$\underset{||u||=1}{\operatorname{argmax}} ||Y^T Xu||^2 = \underset{||u||=1}{\operatorname{argmin}} (||Y - Xu||^2 - ||Xu||^2) \tag{3}$$

$$= \underset{||u||=1}{\operatorname{argmin}} (||Y - Xu||^2 + ||X - Xuu^T||_F^2). \tag{4}$$

---

**Algorithm 1** Solving `sisPCA-linear` using alternating eigendecomposition

---

**input:** Data $X \in \mathbb{R}^{n \times p}$, kernels for $m$ target variables$\{K_{Y_j}, j = 1, ..., m\}$, dimensions of subspaces $\{d_j, j = 1, ..., m\}$, independence regularization $\lambda$.

Randomly initialize and orthonormalize $U_j$ and, optionally, determine the optimal order of subspace updates through one complete update cycle. Assuming with update order $\{1, ..., m\}$.

**repeat**
    $K_{Z_i} \leftarrow X U_i U_i^T X^T$ for $i \in \{1, ..., m\}$;
    **for** $j \in \{1, ..., m\}$ **do**
        $\tilde{K}_j \leftarrow H(K_{Y_j} - \frac{\lambda}{2} \sum_{i=1, i \neq j}^{m} K_{Z_i})H$;
        Update $U_j \leftarrow$ the first $d_j$ eigenvectors of $X^T \tilde{K}_j X$;
        Update $K_{Z_j} \leftarrow X U_j U_j^T X^T$;
    **end for**
**until** converge
**return:** Linear projections $\{U_j\}$ and subspace representations of data $\{Z_j := X U_j\}$ for $i \in \{1, ..., m\}$.

---

That is, maximizing the linear $HSIC(Xu, Y)$ solves the regression problem of approximating the target $Y$ with $u$ serving as the regression coefficients, while also maximizing variations of the fitted value $\hat{Y} := Xu$ (or, equivalently, minimizing an additional self-supervised reconstruction loss). The particular regularization term in (3) corresponds to a zero-mean multivariate Gaussian (MVN) prior on $u$. To see this, we first diagonalize $X^T X$ into $P^T SSP$ where $P$ is orthogonal and $S = \text{diag}\{\sigma_1, ..., \sigma_p\}$ contains the singular values of $X$ in decreasing order. Denote $k := Pu \in \mathbb{R}^{p \times 1}$. Now we have

$$-||Xu||^2 = -(Pu)^T S (Pu) = -\sum_{i=1}^{p} \sigma_i^2 k_i^2 = \sum_{i=2}^{p}(\sigma_1^2 - \sigma_i^2)k_i^2 - \sigma_1^2 ||k||^2 \overset{||k||=1}{=\!=\!=} ||\tilde{X}u||^2 - \sigma_1^2$$

where $\tilde{X} := (\sigma_1 I - S)P$. Subsequently, (3) becomes

$$\underset{||u||=1}{\text{argmin}}(||Y - Xu||^2 + ||\tilde{X}u||^2), \tag{5}$$

and solving (5) is equivalent to finding the maximum a posteriori estimator of the regression coefficients $u$ under a MVN prior with zero mean and inverse covariance $\tilde{X}^T \tilde{X}$. Directions corresponding to smaller singular values will receive stronger shrinkage effects. The formulation of (5) is also closely related to Zellner's g-prior in Bayesian regression where the coefficient $\beta$ follows $\beta \sim MVN(0, g\sigma^2(X^T X)^{-1})$ for some positive $g$ and noise variance $\sigma^2$. Finally, when fixing other subspaces $\{Z_v\}$, $HSIC(Xu, Z_v) := (Xu)^T K_{Z_v}(Xu)$, now a function of $u$, can also be integrated into (5). The new quadratic regularization term is $Q(u) := u^T(X^T K_{Z_v} X + \tilde{X}^T \tilde{X})u$, which again corresponds to a zero-centered MVN prior.

## C  Optimization landscape of sisPCA-linear and Conjecture 3.1

We first provide a motivating example to see how the first supervision term in (2) affects the optimization landscape. Assume no supervision $Y$ (i.e. $K_{Y_m} = I$) and all the latent subspaces $\{Z_m\}$ share the same dimension $d$. In the absence of disentanglement regularization, every subspace will simply be the same vanilla PCA space, leading to complete overlap. The introduction of the disentanglement penalty in (2) maintains the objective's symmetry with respect to $U_m$. In other words, subspaces are interchangeable in the pure unsupervised setting, and consequently the symmetry gives rise to multiple local optima. Nevertheless, the presented challenge is trivial, in that all local optima are also global, and the number of local optima, considered in terms of invariant groups up to column sign-flipping, depends only on the number of subspaces $m$.

Under supervision, some subspaces may be selectively favored. For example, the scaling of subspace supervision kernels $\{K_{Y_m}\}$ can break the symmetry of the overall objective, allowing information to be preferentially preserved in one subspace while being removed from others. In such scenarios (referred to as *unbalanced supervision*), local optima that are closer to the supervised PCA results of

subspaces with larger $K_{Y_m}$ also yield higher objective values. The *balanced supervision condition* can be evaluated by calculating the relative ratio of kernel scales, or equivalently, the ratio of objective gradient with respect to each subspace kernel.

Another important observation is that, because the disentanglement penalty is symmetric, tuning the hyperparameter $\lambda$ will not alter the ranking of local optima. This suggests a straightforward initialization strategy to navigate towards the global solution by evaluating the relative supervision strength, $HSIC(Z_m^0, Y_m)$, for each subspace. Here, $Z_m^0$ represents the initial, pre-disentanglement solution for subspace $m$, which can be efficiently computed using supervised PCA.

We can directly visualize the optimization landscape in a simplified scenario (Fig. 7). Consider $X = \begin{pmatrix} 1 & 0 \\ -1 & 0 \end{pmatrix}$, which consists of two 2D vectors separated along the first axis. With supervisions $Y_1 = (1, -1)^T$ and $Y_2 = cY_1$ differing only by a scalar $c$, our goal is to determine the projections $U = (u, u_2)^T$ and $V = (v, v_2)^T$ that transform $X$ into subspace representations $Z_1 = XU = (u, -u)^T$ and $Z_2 = XV = (v, -v)^T$. Adopting a linear kernel for $K_Y$, the `sisPCA-linear` objective (2) is now equivalent to the following simplified problem

$$\underset{u,v \in [0,1]}{\arg\max} f(u, v) = u + cv - \lambda uv.$$

Since $X$ only has one effective dimension, both $U$ and $V$ align with that axis (i.e., $u = 1$ and $v = 1$, the unregularized solution) when the disentanglement penalty $\lambda$ is low ($\lambda < 1$), leading to a single optimal solution $(u, v) = (1, 1)$. Under strong regularization, two local maxima $(1, 0)$ and $(0, 1)$ emerge, and their relative order depends solely on the scalar $c$ and is unaffected by $\lambda$. Specifically, each maximum represents a preference for capturing the significant axis in $X$ within one subspace, allowing for an easy decision on initialization based on a comparison between $u$ and $cv$. In the context of alternating optimization methods like Algorithm 1, it can be simply done by fixing the subspace $V$ ($v = 1$) and updating $U$ first to minimize the disentanglement penalty (from $u = 1$, the unregularized solution, to $u = 0$). The optimization path stays near the global optima regardless of $\lambda$.

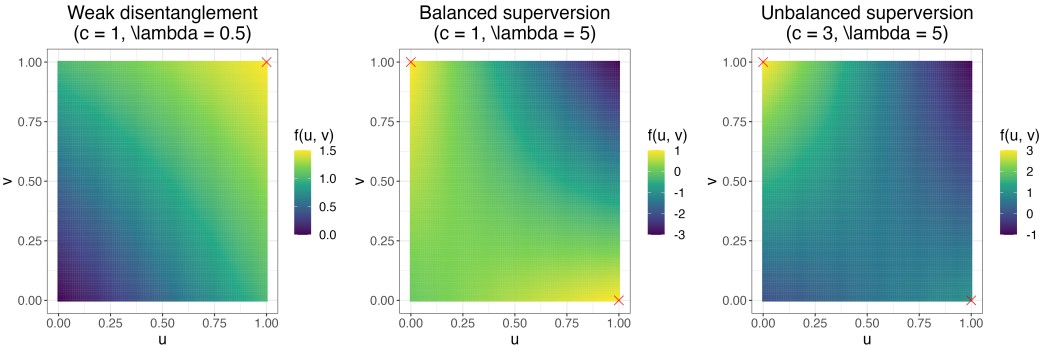

Figure 7: A simplified example of the optimization landscape of `sisPCA-linear`. Under balanced supervision ($c = 1$), the symmetry-induced two local maxima are both global regardless of $\lambda$. When $c \neq 1$, one solution is favored than the other, and the relative order is also independent of $\lambda$.

This phenomenon also resembles the situation described in Ge et al. [2017] on $f(UV^T)$, where an additional regularizer $||U^T U - V^T V||_F^2$ is necessary to maintain balance when dealing with asymmetric matrices. The first half of Conjecture 3.1 is indeed motivated by the above result. For `sisPCA`, a similar regularizer in the form of $||Z_i Z_i^T - Z_i Z_i^T||_F^2$ may be enough to ensure balanced supervision. Moreover, given that the multi-solution challenge arises from subspace interchangeability, we hypothesize that more discriminative $Y$ could alleviate the issue. Since there are only pairwise interactions between subspaces in (2), Ge et al. [2017]'s framework can be adapted to study the behavior of more than two subspaces. Intuitively, the proof of Conjecture 3.1 would be to show that the local gradient direction of (2) aligns with the global descent direction in a region near global optima, and proper regularization or initialization can make sure the optimizer stays in that region. Following the procedures described in section 5 of Ge et al. [2017] for asymmetric matrices, we may concatenate all subspaces into $W := [Z_1, ..., Z_m]^T$ and reformulate the objective as a quadratic function of $N := WW^T$ plus some regularization.

# D  Design and applications of `sisPCA-general` with Gaussian kernel

In this section, we discuss the `sisPCA-general` optimization problem in detail. Recall that it is motivated by the limitation of `sisPCA-linear` that zero HSIC disentanglement regularization does not ensure statistical independence. In `sisPCA-general`, we want to solve (1) by employing HSIC with with *universal* kernels, e.g. the Gaussian kernel.

**Definition D.1.** Given a compact metric space $\mathcal{X}$ and a continuous kernel $K$ on $\mathcal{X}$, $K$ is called *universal* if for every continuous function $g \in C(\mathcal{X})$ and all $\epsilon > 0$, there exists an $f \in \mathcal{H}$ such that $||f - g||_{\mathcal{X}} < \epsilon$.

A nice result on universal kernel from the Constrained Covariance (COCO) framework [Gretton et al., 2005b] is that it can approximate any continuous function on a compact metric space. In other words, the HSIC with a universal kernel will be zero only if, among all possible continuous transformation, the covariance of two random variables are always zero. In `sisPCA-general`, the latent representation is still encoded by a linear projection $Z_j = XU_j$. This indicates that $U_j$ can be interpreted in the same way as PCA and `sisPCA-linear` to extract feature importance.

We solve the `sisPCA-general` problem with Gaussian kernel,

$$k(x, x') = \exp(-w^2||x - x'||^2),$$

using gradient descent (Algorithm 2). Since the problem is constrained, the algorithm unfortunately is not guaranteed to converge. The HSIC with Gaussian kernel also affects the optimization landscape. Empirically, we observed that `sisPCA-general` tends to have many local optima, requiring exhaustive tuning of training parameters such as the learning rate and stopping criteria. The Gaussian kernel scale $w$ is also important. Ma et al. [2020] reported in their experiments that the performance of HSIC-based optimization is moderately sensitive to $w$ and thereby proposed a multiple-scale solution that essentially aggregates results from different $w$. In our experiments, we followed Gretton et al. [2005a] by setting the kernel scale to the median distance between data points in the input space.

Another notable disadvantage of `sisPCA-general` is its cubic-scaling computational complexity. Here we focus on scenarios where the number of features $p$ is fixed and the sample size $n$ is large ($n > p$), which is common when data preprocessing and filtering is in place. Specifically, the computational complexity for calculating $HSIC(X, Y)$ with a linear kernel for $X \in \mathbb{R}^{n \times p}$ and a given target kernel $K_Y$ for $Y$ is $O(n^2 p)$. If both $K_X$ and $K_Y$ are linear, the complexity can be further reduced to $O(np)$. Additionally, the eigendecomposition of the (cross) covariance incurs a cost of $O(p^3)$. This means that each update in Algorithm 1 has an overall complexity of $O(n^2)$ wrt the sample size $n$, aligning with that of supervised PCA. In comparison, most dimensionality reduction methods requiring pair-wise distance computation have a complexity of at least $O(n^2)$. On the other hand, for `sisPCA-general`, computing a Gaussian-kernel $HSIC(X, Y)$ naively is of $O(n^3)$ complexity. From an engineering perspective, further optimizations for both the `sisPCA-linear` (Algorithm 1) and `sisPCA-general` (Algorithm 2) algorithms are conceivable, for instance, by exploiting data sparsity and by mini-batch or incremental update.

---

**Algorithm 2** Solving `sisPCA-general` using gradient descent

---

**input:** Data $X \in \mathbb{R}^{n \times p}$, kernels for $m$ target variables $\{K_{Y_j}, j = 1, ..., m\}$, dimensions of subspaces $\{d_j, j = 1, ..., m\}$, a universal kernel $\Phi_Z(\cdot, \cdot)$ for latent representations, independence regularization $\lambda$.

Randomly initialize and orthonormalize $U_j$ and, optionally, determine the optimal order of subspace updates through one complete update cycle. Assuming with update order $\{1, ..., m\}$.

**repeat**

    $K_{Z_i}(k, r) \leftarrow \Phi_Z(X_k U_i, X_r U_i)$ for $i \in \{1, ..., m\}$;

    $L \leftarrow -\sum_{j=1}^m tr(K_{Z_j} H K_{Y_j} H)$;

    $L \leftarrow L + \lambda \sum_{j=1}^m \sum_{i>j}^m tr(K_{Z_i} H K_{Z_j} H)$;

    Jointly update $\{U_j\}$ by minimizing $L$ using gradient descent;

    Project $U_j$ to the orthonormal space;

**until** converge

**return:** Linear projections $\{U_j\}$ and subspace representations of data $\{Z_j := XU_j\}$ for $i \in \{1, ..., m\}$.

---

**Recovering supervised and unsupervised subspaces in simulated data** We demonstrate the performance and limitation of `sisPCA-general` using simulated data. The task and simulation details are described in Section 4.1. As illustrated in Fig. 8, `sisPCA-general` partially recovers the two supervised subspaces S1 and S2. Challenges arise, however, in fully eliminating continuous S2 signatures from S1. This difficulty may stem from the smaller scale of HSIC values with Gaussian kernels as compared to those with linear kernels, and a more complicated optimization landscape due to the non-linearity introduced in the regularization. Additionally, the unsupervised subspace S3 appears to collapse to one dimension, which partially illustrate the identifiability issue raised in Section 3.3. The issue may be more pronounced in `sisPCA-general` because of the diminished HSIC scale (i.e. weaker connection between subspaces and their target variables).

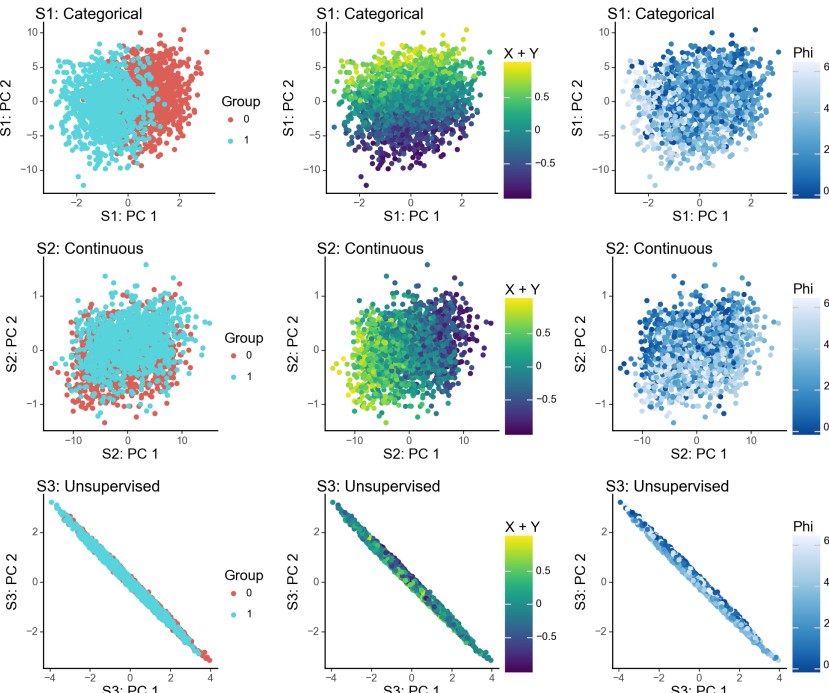

Figure 8: Performance of `sisPCA-general` ($\lambda = 30$) on the simulated dataset. Related to Fig. 3.

# E    Extending sisPCA to nonlinear settings using kernel transformation

Following kernel PCA's methodology [Schölkopf et al., 1997], we can extend `sisPCA` for nonlinear scenarios where features in $X$ interact through a feature map $\Phi : \mathbb{R}^p \to F$. The space $F$ may have infinite dimension, and we can only explicitly write out the projection $U$ in the finite case $F \subset \mathbb{R}^k$, for example, $\Phi(x) = x^2$. In that context, the latent representation $Z := \Phi(X)U$ can be naively computed, and the `sisPCA` objective (1) will stay the same. By simply replacing $X$ with $\Phi(X)$, methods such as gradient descent and Algorithm 1 still hold. However, the beauty of the *kernel trick* is that we can bypass the computation of $\Phi(X)$ to obtain $Z$. Instead of learning the projection matrix $U$, which is no longer possible when $F$ is infinite-dimensional, we can learn a set of coefficients $\vec{\alpha}$ from

$$Z = \Phi(X)U := \sum_{i}^{n} \vec{\alpha}_i k(X, X_i),$$

with $\{X_i\}$ representing the $n$ data points and $k(x, y) := \langle \Phi(x), \Phi(y) \rangle$ the corresponding kernel of $\Phi$. Note that $k(\cdot, \cdot)$ is specified wrt to *the data* $X$, which is different from the kernel $K_z$ for latent space and the kernel $K_y$ for supervision. We leave the practical implementation of this extension to future exploration.

Table 4: General comparison of baseline models.

| | Linear | | | Non-linear | | |
|---|---|---|---|---|---|---|
| | PCA | sPCA | **sisPCA** | VAE | supVAE | hsVAE |
| Supervision | - | HSIC | HSIC | - | Prediction | Prediction |
| Disentanglement | - | - | HSIC | - | - | HSIC |
| Interpretability | | $U$ as feature importance | | | Black-box | |
| Hyperparameters | (1) | (1,2) | (1,2,3) | [1,2] | [1,2,3] | [1,2,3,4] |

<table>
<tr><td>(1) Subspace dimension</td><td>[1] Subspace dimension</td></tr>
<tr><td>(2) Subspace kernel</td><td>[2] Autoencoder design</td></tr>
<tr><td>(3) HSIC penalty strength $\lambda$</td><td>[3] Predictor design</td></tr>
<tr><td></td><td>[4] HSIC penalty strength $\lambda$</td></tr>
</table>

| Optimization | Closed form | Simple | Subject to general training limitations of deep generative models[*] |
|---|---|---|---|

[*]VAEs could not be run on the 6-dimensional simulated data in Figure 3 due to NaNs generated during variational training. This is a common issue in training SCVI models likely resulting from parameter explosion.

## F   Baseline section and experiment details

### F.1   Overall evaluation criteria

We evaluate model performance based on two key aspects:

**Representation Quality:** This criterion assesses the ability to learn low-dimensional representations that reflect specific and unmixed data properties. A high-quality subspace representation should contain minimal confounding information from other subspaces. We employ both quantitative metrics (e.g., information density and subspace separateness) and qualitative 2D visualizations of learned representations.

**Interpretability:** This criterion evaluates the ease of understanding feature contributions to different data properties. For linear methods (PCA, sPCA, and sisPCA), we consider the learned projection $U$ as feature importance and extract top features from the loading (see examples in Sections 4.2 and 4.4). Non-linear models generally lack straightforward feature-to-subspace mapping, making them inherently less interpretable. While approaches like gradient-based saliency maps are becoming standard for interpreting black-box models such as VAEs, they are less straightforward and face challenges in aggregating sample-level gradients into global feature importance scores. Therefore, we only compare sisPCA to other linear models for interpretability analysis (e.g. ranking and selecting top genes according to the loading).

### F.2   Baseline model design and general comparison

Table 4 provides a comprehensive comparison of linear and non-linear models used in this study, including the linear PCA, sPCA and sisPCA (proposed work) described in the Methods section, as well as sevral non-linear VAE counterparts inspired by the work of HCV [Lopez et al., 2018]:

- **VAE:** Vanilla VAE with Gaussian likelihood.
- **supVAE:** Gaussian VAE with additional predictors for target variables.
- **hsVAE:** supVAE with additional HSIC disentanglement penalty (Gaussian kernel). It is essentially a general-purpose HCV [Lopez et al., 2018] with Gaussian likelihood.

In the sisPCA package[5], PCA and sPCA are implemented as special cases of sisPCA and solved analytically using eigendecomposition. VAE models and variational inference are implemented using the SCVI framework[6]. The VAE model is a special case of SCVI's VAE module (latent_distribution = "normal", dispersion = "gene") but with a Gaussian generative model. The supVAE is a multi-subspace extension of VAE with additional predictors to predict target supervisions from the corresponding

---

[5]https://github.com/JiayuSuPKU/sispca
[6]scvi-tools v1.2.0 (https://scvi-tools.org/)

latent representations. The last layer of the predictor is either a linear transformation for continuous supervision or linear followed by softmax for categorical supervision. The hsVAE model is an extension of supVAE where we add additional subspace disentanglement penalty (i.e., HSIC with a Gaussian kernel) to the objective, as described in HCV [Lopez et al., 2018].

For single-cell RNA-seq data, we further implement a domain-specific hsVAE model, namely hsVAE-sc. Specifically, hsVAE-sc uses negative binomial as the data generative distribution with an extra autoencoder to learn the library size (the sum of total counts per cell) and is correspondingly trained on counts-level data. It is very close to a re-implementation of the scVIGenQCModel developed in the HCV paper under the latest SCVI framework.

### F.3    Hyperparameter selection

As outlined in Table 4, `sisPCA` has three main hyperparameters:

1. Subspace latent dimension $d$.
2. Subspace supervision kernel $K_Y$.
3. Disentanglement penalty strength $\lambda$.

The tuning of (3) will be discussed in Appendix G. Here we illustrate the selection of (1) and (2), which are largely pre-determined based on the target variable supplied for supervision.

In our analysis, we consistently use the delta kernel for categorical targets and the linear kernel for continuous targets (Section 3.2). Given a supervision kernel $K_Y$ of rank $k$, the *effective dimension* of the corresponding `sisPCA` subspace, defined as the number of dimensions with non-zero variance, is determined by the eigendecomposition step in Algorithm 1. Specifically, we have the following upper bounds:

- For sPCA ($\lambda = 0$): The effective dimension equals $\text{rank}(X^T K_Y X) \leq \min(k, p) \approx k$, where $p$ is the number of features in $X$.
- For `sisPCA`: The effective dimension equals $\text{rank}(X^T \tilde{K}_Y X) \leq \min(k + d, p) \approx k + d$, where $\tilde{K}_Y \leftarrow K_Y - \lambda Z^T Z$ is the supervision kernel after the rank-d disentanglement update and $d$ is the number of latent dimensions of $Z$.

In practice, we observe that the effective dimensions of both sPCA and `sisPCA` closely approximate the target kernel rank $k$. For instance:

- For a 1-D continuous variable with linear kernel (e.g., age in Section 4.3), the effective dimension is always 1.
- For a categorical variable of $k$ groups with delta kernel (e.g., time point in Section 4.4), the effective dimension is approximately $k$.

Consequently, we find that the learned `sisPCA` subspaces remain consistent regardless of the specified dimension. Any additional axes beyond the effective dimension tend to collapse to zero and are primarily affected by numeric errors in the SVD solver. Since `sisPCA` orders subspace dimensions by the variance explained (i.e., eigenvalues), it is straightforward to remove these extra dimensions by examining the variance explained curve.

In contrast, VAE models are more sensitive to dimension changes and generally have more hyperparameters to tune (Table 4). Designing optimal architectures for autoencoders and target predictors is a topic in itself and beyond the scope of our current work. For benchmarking purposes, we largely follow the default VAE architecture from SCVI:

- Autoencoders for latent mean and variance: One hidden layer with 128 hidden units, ReLU activation, batch normalization, and dropout.
- Predictor design (adapted from the scVIGenQCModel in the HCV paper): One hidden layer with 25 neurons, ReLU activation, and dropout. Extra softmax output head for classifier.
- Training objective: Equal weighting of prediction error and reconstruction loss, minimized using variational inference.

In Section 4.4, we set the subspace dimension to 10 for all VAE models. We use UMAP to project the learned subspace onto 2D and subsequently calculate the Silhouette score in Table 3 on this 2D subspace, ensuring dimensional consistency with the PCA-based spaces (which are also projected onto 2D using UMAP).

### F.4 Additional experiment details

**Recovering Supervised and Unsupervised Subspaces in Simulated Data** The code used to simulate the donut data in Section 4.1 is available in the `sisPCA` GitHub repository. We have verified that altering the noise distribution (e.g., from Gaussian to uniform noise, or from uniform projection matrix to Gaussian matrix) yields similar results.

**Learning Diagnostic Subspaces from Breast Cancer Image Features** We obtained the dataset from Kaggle and scaled each real-valued feature to have zero mean and unit variance. The input for PCA includes all 30 quantitative features. The 'radius' subspace is supervised with 'radius_mean' and 'radius_sd' (effective dim = 2), while the 'symmetry' space is supervised with 'symmetry_mean' and 'symmetry_sd' (effective dim = 2). For sPCA and `sisPCA`, we use the remaining 26 features as input and project the data onto each subspace. Using the elbow method, we set the PCA subspace dimension to 6, and correspondingly, the dimension for sPCA and `sisPCA` subspaces to 3 (although their effective dimension is approximately 2). The Silhouette scores are computed on the first three principal components for all subspaces.

**Separating aging-dependent DNA methylation changes from tumorigenic signatures** We downloaded the TCGA methylation data from GDC (controlled access) and retained only the first 5,000 non-constant and non-NA CpG sites for analysis. The data was zero-centered. All subspaces are specified with 10 latent dimensions. However, as noted in the previous section, the sPCA and `sisPCA` aging subspaces have an effective dimension of only one.

**Disentangling infection-induced changes in the mouse single-cell atlas of the *Plasmodium* liver stage** We used the single-cell data preprocessed by Piran et al. [2024][7]. The processed data comprises 19,053 cells and 8,203 mouse genes (all malaria genes have been filtered out), and we keep the top 2,000 variable genes using scanpy's 'sc.pp.highly_variable_genes' function. For all models except hsVAE-sc, we use the normalized and log1p-transformed expression data as inputs. For hsVAE-sc, we supply the raw data before normalization. It's important to note that the data underwent a background correction step before normalization, where the mean expression in empty wells was subtracted from the observed data. This results in the raw counts being technically non-integers. Consequently, in hsVAE-sc, a continuous extension of the negative binomial likelihood is used to compute the reconstruction loss. All subspaces are given 10 latent dimensions. Although based on the kernel rank, we know that the sPCA and `sisPCA` infection subspaces have an effective dimension of 2, while the time subspaces have an effective dimension of 6.

## G Tuning the disentanglement strength in sisPCA

We adapt the approach from Abid et al. [2018] to develop a computational pipeline for automated selection of the `sisPCA` hyperparameter $\lambda$. The process involves:

1. Fitting `sisPCA` models with multiple $\lambda$ values.
2. Computing pairwise similarity (affinity) between learned subspaces.
3. Identifying representative subspace clusters using spectral clustering.

We compute subspace affinity based on the principal angle [Miao and Ben-Israel, 1992]:

$$d_{subspace}(Z_1, Z_2) := d(\{\theta_1, ..., \theta_k\}) := d(\Theta)$$

where $Z \in \mathcal{R}^{n \times k}$ is the orthonormal basis of a dim-k subspace with $n$ observations, and the principal angles $\Theta$ are computed from the SVD of $Z_1^T Z_2 = U \cos(\Theta) V^T$.

Through empirical testing, we found the Fubini-Study Grassmann distance provides optimal performance. Thus, we define subspace affinity as:

$$\text{Affinity}_{subspace}(Z_1, Z_2) = \prod_{i=1}^{k} \cos(\theta_i) = \prod_{i=1}^{k} \sigma_i(Z_1^T Z_2).$$

---

[7]See the notebook https://github.com/nitzanlab/biolord_reproducibility/blob/main/notebooks/spatio-temporal-infection/1_spatio-temporal-infection_preprocessing.ipynb

For stability, we truncate subspace latent dimension to $k' := \min(\text{rank}(K_Y), k)$, where $K_Y$ is the corresponding supervision kernel (see discussion on subspace effective dimension in Appendix F.3). For `sisPCA` models with multiple subspaces, we compute model-wise affinity as the average of subspace affinities

$$\text{Affinity}_{model}(\{A_1, ..., A_m\}, \{B_1, ..., B_m\}) = \frac{1}{m} \sum_{i=1}^{m} \text{Affinity}_{subspace}(A_i, B_i).$$

**Learning Diagnostic Subspaces from Breast Cancer Image Features**   We evaluated 20 $\lambda$ values ranging from 0 to 100 and analyzed the pairwise similarity between learned `sisPCA` subspaces (Fig. 9). The sPCA solution ($\lambda = 0$) shows clear separation from other `sisPCA` models. As $\lambda$ increases, the symmetry subspace becomes progressively less predictive of diagnostic status (Fig. 9b), and subspaces stabilize and converge to a robust solution after $\lambda = 1$. This convergence pattern is also reflected in the elbow of the reconstruction loss curve (Fig. 9c).

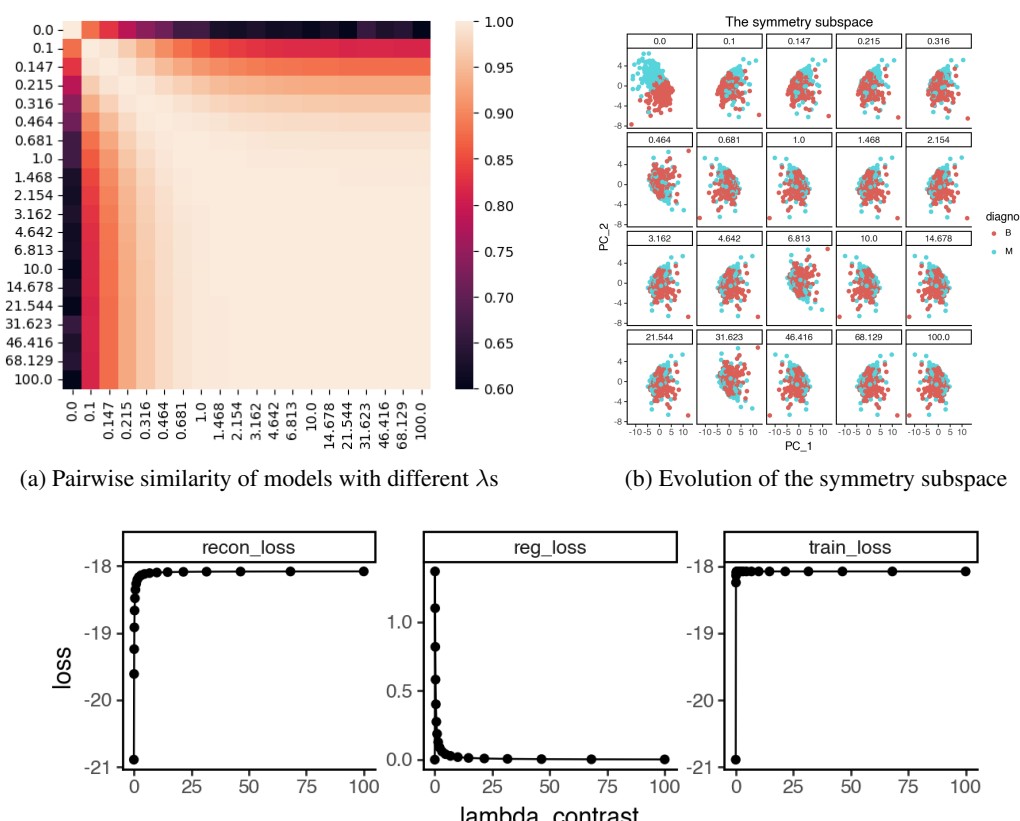

(a) Pairwise similarity of models with different $\lambda$s         (b) Evolution of the symmetry subspace

(c) Training loss as functions of $\lambda$ (total loss = reconstruction loss + disentanglement regularization)

Figure 9: Effect of $\lambda$ on the learned subspace structure in the breast cancer dataset. Related to Fig. 4.

**Disentangling infection-induced changes in the mouse single-cell atlas of the *Plasmodium* liver stage**   We examined 10 $\lambda$ values ranging from 0 to 100 to assess their effect on subspace characteristics (Fig. 10). Our analysis again reveals the clear separation of sPCA solution ($\lambda = 0$) from other `sisPCA` models. As $\lambda$ increases, infected cells form a tighter cluster in the infection subspace (Fig. 10a), while temporal dynamics become less pronounced, with increased mixing of cells from different time points (Fig. 10b). Moreover, `sisPCA` results are quite robust across $\lambda$ in preserving subspace structure (Fig. 10c). Since the disentanglement effect primarily manifests in PC2 of the infection subspace, all models, even sPCA, extract similar sets of top contributor genes in PC1 of the infection subspace, thus yielding GO enrichment results comparable to those shown in Fig. 6.

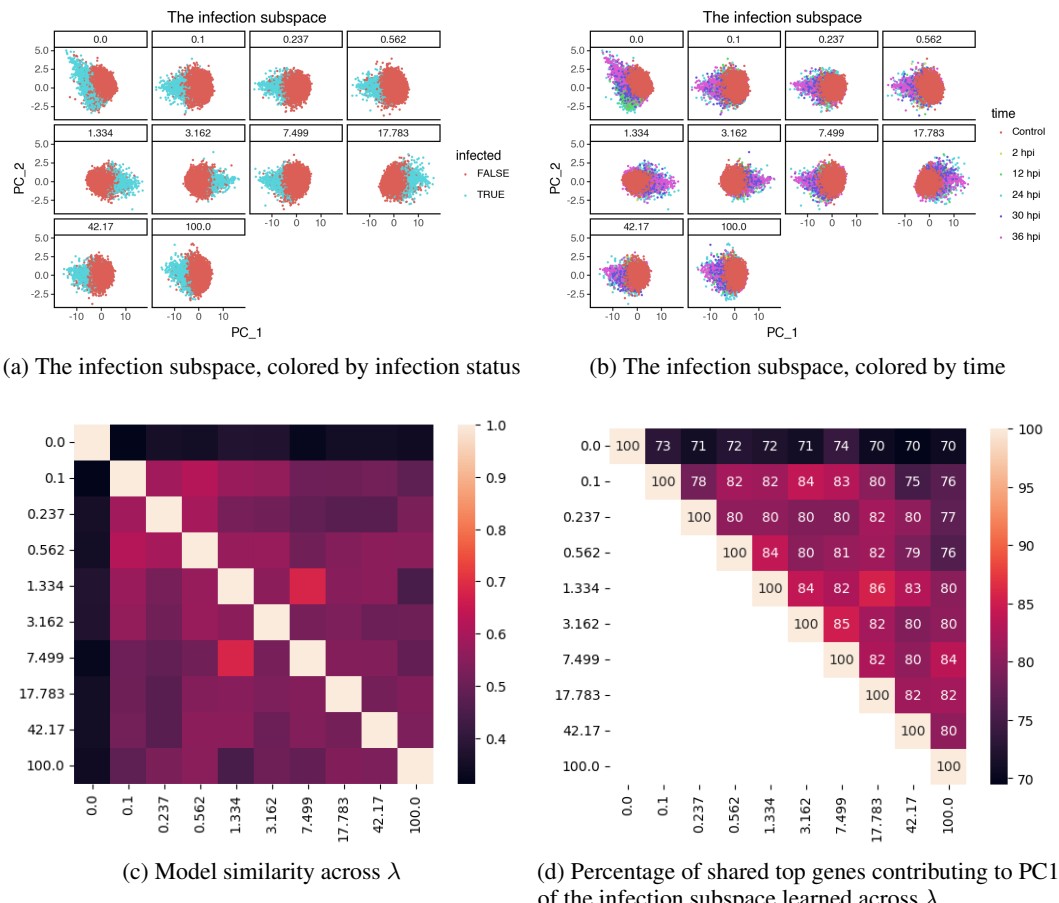

(a) The infection subspace, colored by infection status

(b) The infection subspace, colored by time

(c) Model similarity across $\lambda$

(d) Percentage of shared top genes contributing to PC1 of the infection subspace learned across $\lambda$

Figure 10: Effect of $\lambda$ on the learned subspace structure in the single-cell malaria infection data. Related to Fig. 5 and Fig. 6.

## H    Supplementary visualizations of baseline performance in applications.

For completeness, here we provide subspace visualization for baseline models not included in the main figures. Specifically, Fig. 11 shows the full sPCA performance on the simulated data, related to Fig. 3. Fig. 12 visualizes the single-cell subspaces learned by sPCA, VAE, supVAE, and hsVAE-sc, related to Fig. 5.

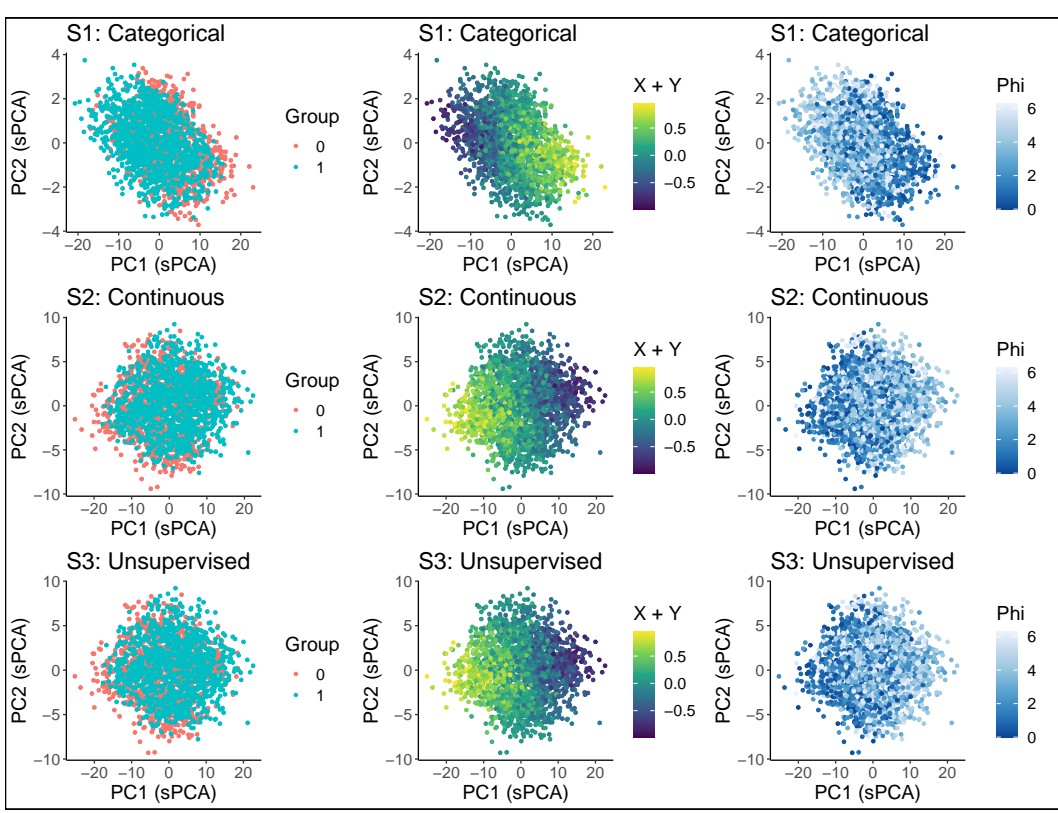

Figure 11: sPCA results on the simulated dataset. Related to Fig. 3b. Rows from top to bottom: sPCA results with S1 categorical supervision; with S2 continuous supervision; without supervision (vanilla PCA). In this case, all sPCA subspaces are dominated by S2 since it carries the strongest variability.

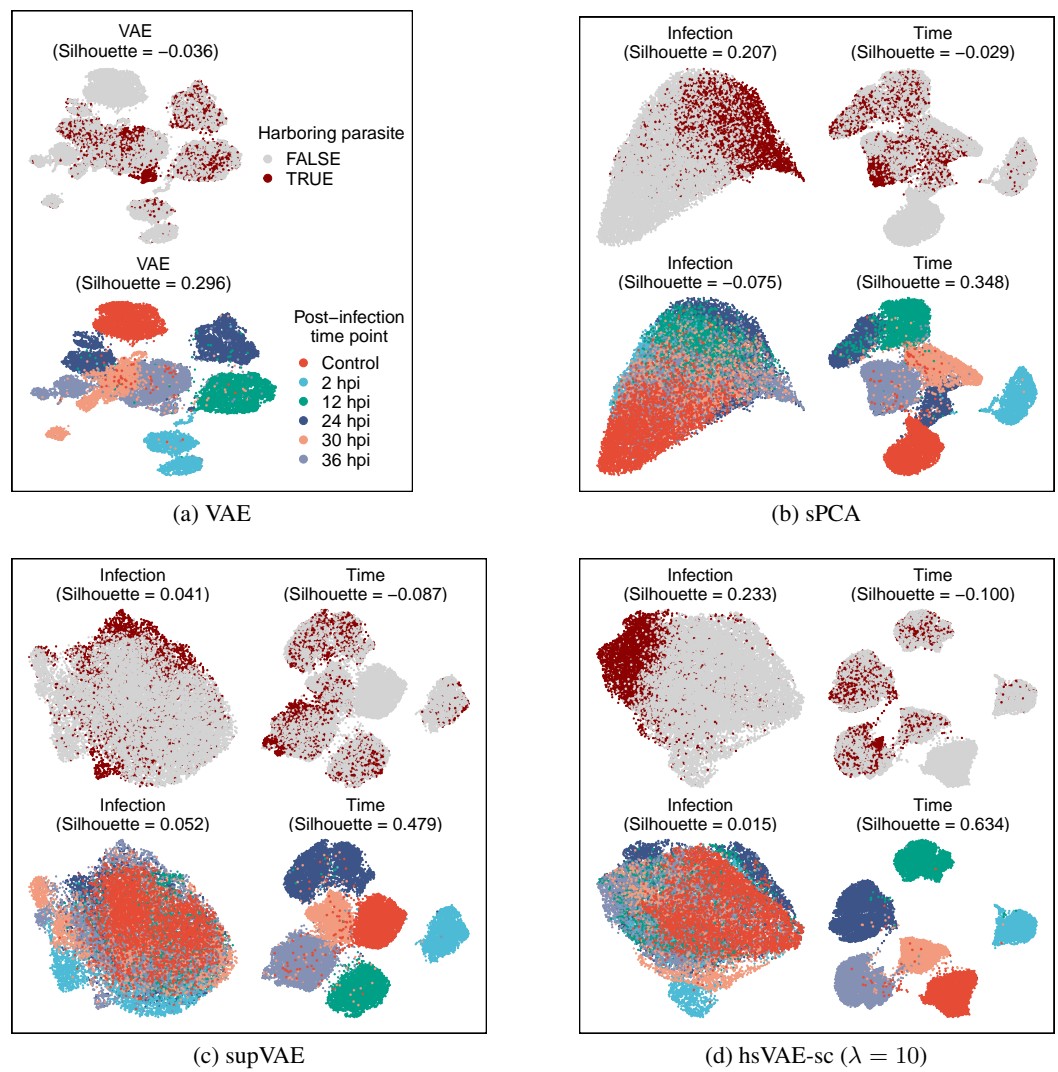

Figure 12: UMAP visualizations of the scRNA-seq data of mouse liver upon *Plasmodium* infection. Subspace representations are learned using unsupervised VAE (a) and supervised sPCA (b), supVAE (c) and hsVAE-sc (d). Note that the infection subspaces of VAE and supVAE fail to distinguish infected versus uninfected cells. Moreover, all infection subspaces presented here still exhibit significant temporal patterns (lower left plot in each panel) where cells collected at different time points are not fully mixed.

