# OpenReview forum: "Disentangling Interpretable Factors with Supervised Independent Subspace Principal Component Analysis"
_NeurIPS.cc/2024/Conference — NeurIPS 2024 poster_

### Official Review · Reviewer_debZ · 2024-07-03

**Soundness:** 3
**Presentation:** 3
**Contribution:** 3
**Rating:** 7
**Confidence:** 4

**Summary:**

Here the authors propose an extension of PCA that decomposes datasets into multiple independent subspaces that are encouraged to reflect provided covariates of interest. To enforce these dependencies, the authors leverage the Hilbert-Schmidt Independence Criterion (HSIC).

**Strengths:**

In short, my views of the manuscript are generally positive. Some specific thoughts below:

* **Clarity**: I found the manuscript well written and easy to follow. Well done!
* **Motivation**: A recent line of work [1,2] has observed that disentangling separate sources of variation in case-control datasets (e.g. in scRNA-seq studies [3-4]) can assist with data exploration tasks. This work extends similar ideas to settings where no explicit case versus control datasets exist, but samples can still be associated with different covariates of interest. I can imagine this work will similarly prove useful in a number of data exploration tasks.
* **Novelty**: To my knowledge, the authors proposed method is indeed novel, and seems like a sensible extension of the previous supervised PCA method and recent works on supervised disentanglement (e.g. [1-2]).

[1] Abid et al., "Contrastive principal component analysis"
[2] Abid et al., "Contrastive variational autoencoder enhances salient features"
[3] Weinberger et al., "Isolating salient variations of interest in single-cell data with contrastiveVI"
[4] Jones et al., "Contrastive latent variable modeling with application to case-control sequencing experiments"

**Weaknesses:**

While my views of the manuscript are generally positive, I do have a couple of minor concerns. If the authors are able to address my concerns I would be happy to raise my score.

* **Impact of $\lambda$**: The authors' method attempts to find a balance between encouraging dependence between subspaces and their corresponding covariates while also minimizing the dependence between individual subspaces, and this balance is tuned via $\lambda$ in Equation (1). I imagine this parameter could have a significant impact on the embeddings/loadings produced by sisPCA, though this isn't explored in the manuscript. It would be great if the authors could perform additional experiments exploring these changes. For example, do the GO results presented in Figure 5 vary for different values of $\lambda$? How different do the UMAPs look for the scRNA-seq data for different values of $\lambda$?

* **Automatic selection of $\lambda**: Related to my previous point, the authors briefly mention that $\lambda$ could be selected using a procedure similar to [1]. It would be great to see an example of this in practice: forcing the user to select $\lambda$ manually seems like it could present a big obstacle in terms of usability, so I think it would be useful to see how the automated method performs.

* **Re: connections with self-supervised learning**: In the abstract the authors note "We elucidate the mathematical connections of
sisPCA with self-supervised learning and regularized linear regression". While I was able to locate the regularized linear regression results in Appendix B.1, I couldn't find any additional mathematical details on the self-supervision connection (the only other mention of self-supervision in the main text was in line 83 in the related works section). Could the authors expand upon this connection? (Please let me know if I missed anything here).

Minor points (these did not affect my score):

* It appears that the authors may have used \citet accidentally instead of \citep throughout the manuscript.
* Figure quality was relatively low-resolution when I printed out the manuscript- perhaps the authors could try increasing the DPI?
* In line 91 the authors mention "contrastive learning"- it may be worth putting explicit citations to e.g. Abid et al. [1] to clarify for the reader that "contrastive learning" here isn't referring to contrastive in the sense of e.g. SimCLR.
* I couldn't find any details on how the real-world biological datasets were preprocessed (e.g. I assume library size normalization was applied to the scRNA-seq data?)

[1] Abid et al., "Contrastive principal component analysis"

**Questions:**

Did the authors try applying their method with different values of $\lambda$ to the real-world datasets?

**Limitations:**

The greatest limitations I see are (1) the method relying on linear transformations (as opposed to e.g. deep neural networks) and (2) potential difficulties in setting $\lambda$. For (1) the authors specifically note that they are intentionally trading off expressivity for increased interpretability

---

> ### Author Rebuttal · Authors · 2024-08-07
>
> ## **Response to Reviewer debZ**
>
> Thank you for your positive feedback and insightful comments! We appreciate your thorough review and would very much like to address any remaining concern.
>
> ### **W1. Impact of $\lambda$ selection and an auto-selection pipeline**
> Thank you for the suggestion. We conducted additional experiments by varying $\lambda$ for the scRNA-seq dataset with values [0, 0.1, 0.3, 1, 3, 10, 30, 100]. On top of that, we implemented a preliminary pipeline for automatic $\lambda$ selection that aggregate resulting subspaces using clustering and outputs aggregated results at different resolution, as implemented in Contrastive PCA. However, due to time constraints, we have yet to thoroughly test some design choices, such as the representation distance, which appears to be important in subspace clustering. To speed up training, we also introduced a mini-batch approach of Alg 1 (Appendix B.2) since the learnable projection matrix *U* is of size (n_feature, n_dim), independent of n_sample. We plan to implement additional features such as parallel training and better initialization search in the future.
>
> Despite these limitations, we confirmed the following results:
>
> #### **Impact on subspace similarity**
> Subspaces learned with similar $\lambda$ values are generally more similar, as measured by Grassmann distance between representations, indicating gradual and continuous changes as a function of $\lambda$. The infection subspaces for $\lambda$ = [0.1, 0.3, 1] cluster together, as do the time subspaces for $\lambda$ = [0, 0.1], while the other subspaces appear more detached.
>
> #### **Impact on gene selection and GO analysis**
> We quantified overlaps of the top 100 genes list for the infection subspace (ranked by absolute PC1 weights) across different $\lambda$ values. Results for $\lambda$ = [0, 0.1, 0.3, 3, 10] are highly consistent, sharing more than 90 of the top 100 genes. The result also suggests that sPCA ($\lambda$ = 0) is sufficient for the GO analysis in Figure 5 ($\lambda$ =10). We examined sPCA representations and confirmed that the confounding effects from the temporal subspace (as visualized in Figure 9b and the new Fig S1c in **M2**) are mainly concentrated in PC2, which is not used here for feature selection. Notably, for reasons still under investigation, $\lambda$ = 1 is an outlier, sharing only 68 of the top 100 genes with sPCA. A possible explanation is that Alg 1 (Appendix B.2) on large datasets may experience numerical issues with PyTorch’s SVD solver.
>
> We will add visualizations of these results in the final version.
>
> ### **W2. Connection with self-supervised learning**
> We apologize for any ambiguity. Our reference of "self-supervised learning" is the connection to the auto-encoder (Appendix B.1), where target variables - features from the data - are used to disentangle the data itself. We chose the term mainly to reflect the difference with supervised learning. Here our goal isn't perfect target prediction, as labels are already known. Instead, we aim to better explain the data by reweighting the self-reconstruction loss with supervision guidance (as reflected in line 401). We recognize the potential confusion and will clarify the term in the final version. The connection to regularized linear regression is discussed in Appendix B.3.
>
> ### **Minor points**
> Thank you for pointing out the problems.
> - We will fix citation issues, and will increase the DPI of all figures to improve resolution.
> - We will provide more details on the preprocessing of the single-cell infection data. The data, preprocessed by the original authors, was subject to library size normalization followed by log1p transformation. A background correction step was also applied before normalization, where the mean expression in empty wells was subtracted. This makes the raw counts technically not integers, but the negative binomial likelihood in hsVAE-sc (**see M2**) seems to work just as well.

---

> > ### Comment · Reviewer_debZ · 2024-08-11
> >
> > Dear authors,
> >
> > Thank you for your thoughtful responses! I've accordingly raised my score.
> >
> > Based on the rebuttal I have a few remaining minor comments:
> >
> > * I would encourage the authors to avoid the name "sVAE" when describing the supervised VAE baseline, as this name has previously been used for a different class of models [1,2]
> > * Two recent works [3,4] have investigated using HSIC-based penalties with VAEs for similar supervised disentanglement tasks; for completeness it would be great if the authors could include citations to these works
> > * Please do include your final results re: the automatic selection procedure for $\lambda$, as I believe this will make a big impact in terms of usability of the method.
> >
> > [1]: Lachapelle et al. "Disentanglement via mechanism sparsity regularization: A new principle for nonlinear ICA"
> > [2]: Lopez et al. "Learning Causal Representations of Single Cells via Sparse Mechanism Shift Modeling"
> > [3]: Tu et al. "A Supervised Contrastive Framework for Learning Disentangled Representations of Cell Perturbation Data"
> > [4]: Qiu et al. "Isolating structured salient variations in single-cell transcriptomic data with StrastiveVI"

---

> > > ### Author Response · Authors · 2024-08-13
> > > **Thank you**
> > >
> > > Thank you for your positive feedback. We very much appreciate your recognitions and all your helpful suggestions for improving our paper. We will include the automatic selection procedure as part of the package release. To further enhance usability, we will also release the re-implemented HCV (hsVAE-sc) models for single-cell data and provide corresponding tutorials on example datsets and general documentations.

---

### Official Review · Reviewer_QSK9 · 2024-07-13

**Soundness:** 3
**Presentation:** 3
**Contribution:** 2
**Rating:** 6
**Confidence:** 4

**Summary:**

This paper proposed a linear dimensionality reduction and subspace extraction method based on Hilbert-Schmidt Independence Criterion (HSIC) and Supervised PCA. In specific, several interpretable subspaces, which are independent from each other, are disentangled from the data observations and the leaned subspaces are enforced to be maximally dependent on the supervision variables. The proposed approach is shown to be effective on both synthetic data and two real datasets.

**Strengths:**

1.	A new method for subspace learning is proposed using supervised PCA and dependence maximization and minimization with HSIC for extracting independent latent low-dimensional subspaces. The framework is corroborated effective in extracting linearly mixed simulation data, and the learned representations are more interpretable than that of supervised PCA.
2.	The problem formulation and optimization are clearly derived and delivered. The overall manuscript is written in an organized way and the notations are well-defined and relatively easy to follow.

**Weaknesses:**

1.	It is not clear if the proposed method is guaranteed to extract the latent subspaces and/or under what conditions/assumptions the method would work or fail. For example, in the synthetic experiment, the mixing matrix is uniformly drawn from [0, 1]. Does it still work if the matrix is changed to Gaussian? Overall, the theoretical analysis is somewhat lacking, e.g., identifiability analysis for linearly mixed subspaces.
2.	Baselines seem missing in the real data applications. Only PCA and supervised PCA are considered here. The authors are encouraged to include more state-of-the-art methods to showcase the superiority of the proposed method, e.g., how does PCA work under such a setting? Are other HSIC-based nonlinear methods, e.g., [1], capable of extracting the independent subspace with interpretability?

[1] Lopez, Romain, Jeffrey Regier, Michael I. Jordan, and Nir Yosef. "Information constraints on auto-encoding variational bayes." Advances in neural information processing systems 31 (2018).

**Questions:**

1.	In Sec. 3.2, the linear kernel is used to target variables. Does it make a difference if a nonlinear kernel function is used?
2.	What are A and B in Fig. 2? Are they the target variables? In the same figure, why does it maximize dependence (HSIC) with the components of a subspace? I thought the only two objectives are 1) maximizing dependence between the subspace and its corresponding target variable; 2) minimizing the dependence between the subspaces. How is this maximization reflected in Eq. (2)?
3.	What does balanced/unbalanced supervision mean mathematically in Conjecture 3.1?

**Limitations:**

The authors are encouraged to discuss limitation of the work in the main paper.

---

> ### Author Rebuttal · Authors · 2024-08-07
>
> ## **Response to Reviewer QSK9**
>
> Thank you for your thoughtful review of our work. We address your main points below:
>
> ## **Weaknesses**
> ### **W1: Theoretical analysis and subspace recovery**
> We acknowledge the limitations in our theoretical analysis. However, sisPCA makes minimal assumptions about the data, and its objective function provides straightforward insights into the model's behavior and optimization landscape (Conjecture 3.1 and Appendix D).
>
> #### **Regarding the simulation example**
>
> 1.	We confirmed that PCA-based models yield similar results with Gaussian-drawn mixing matrices compared to the uniform case. This is because the mixing matrix was normalized to ensure equal contribution from each subspace.
> 2.	As discussed in Appendix D (lines 449-453), subspace scale and supervision strength are the major factors influencing the optimization landscape. For example, sisPCA will prioritize S2’s structure and ignore other subspaces if we scale up the S2 supervision (X, Y) large enough.
>
> #### **General subspace recoverability depends on supervision quality**
>
> 1.	In the most intertwined case of identical supervision for two subspaces, only one can be recovered and the other will collapse to rank zero (Figure 6, Appendix D). Here, linearity ensures that the same information is not split between subspaces - it's an all-or-nothing scenario.
> 2.	When both supervised and unsupervised subspaces are presented, their dynamics are complicated by the dual role of HSIC supervision indicated in Eq.4, Appendix B.3. In particular, the supervised subspace will also try to expand upon the direction that maximize the (unsupervised) variance, potentially leading to identifiability or multi-optima issues (Section 3.3, lines 190-193, Figure 7). Though we reason that the same information will again concentrate in one subspace due to linearity.
> ### **W2: Comparison with state-of-the-art baselines**
> We appreciate your suggestion to include more state-of-the-art methods in our comparison. In response:
>
> 1.	We've added comparisons with non-linear VAE-based models (**M2**), focusing on the real single-cell data where the practical need is to learn genes responsible for malaria infection defense (a feature selection task).
> 2.	Our analysis shows that while VAE-based models can extract independent subspaces, they lack a straightforward method for identifying genes upon which each subspace was constructed (**M2.2**).
> 3.	Regarding representation quality, we've confirmed that linear models like sPCA serve as strong baselines, both qualitatively and quantitatively (**see Fig S1, Table S2 in M2.3**). Specifically, quality of the linear infection subspace often surpasses its non-linear counterpart.
>
> ## **General questions**
> ### **Q1: Effect of non-linear kernels for target variables**
> Yes, the choice of target kernel does influence sisPCA's results, primarily affecting the rank of the learned subspace. This is because sPCA's subspace dimensions are determined by supervision kernel rank (Appendix B.3, line 406-408), and sisPCA shows similar behavior. Specifically, for a linear kernel on *K* continuous features, the effective dimension of the corresponding sisPCA subspace is the rank of the sum of a rank-K kernel and a rank-D disentanglement kernel, where *D* depends on other subspace dimensions (line 146-147). A non-linear target kernel can effectively expand the learned subspace while maintaining interpretability, albeit losing the connection with regularized linear regression (Appendix B.3).
>
> ### **Q2: Clarification on Fig. 2 and maximization objectives**
> We apologize for the confusion. To clarify:
>
> - A and B represent different target variables used for supervision.
> - The maximization arrow within each subspace represents a PCA-like objective of variance maximization, which is equivalent to maximizing the HSIC towards an identity kernel (Appendix B.1, line 397-398).
> - In supervised PCAs, the variance maximization objective comes from the HSIC to the target (first component of Eq.2, see Eq.4 in Appendix B.3). Upon decomposition, the arrows between the subspace and A/B indeed represents minimizing the prediction error (first component of Eq.4), and the arrows within each subspace represents maximizing the variance (second component of Eq.4).
>
> We will revise the figure to make notation consistent.
>
> ### **Q3: Definition of balanced/unbalanced supervision**
> Balanced/unbalanced supervision refers to the relative ratio of scale/norm between target kernels (discussed in Appendix D, line 449-473). It can alternatively be defined as the ratio of loss gradient with respect to each supervision kernel. Intuitively speaking, unbalanced scenarios occur when one kernel is scaled up, favoring the corresponding subspace and creating a single global optimum (Figure 6 in Appendix D).

---

> > ### Comment · Reviewer_QSK9 · 2024-08-11
> >
> > Thank you for the clarification and for including the non-linear VAE-based baselines, which should make this work more solid. I agree that linear methods have more benefits in terms of interpretability and simplicity.

---

> > > ### Author Response · Authors · 2024-08-13
> > > **Thank you**
> > >
> > > Thank you for your feedback. We appreciate your recognition and the suggestion on adding the non-linear baselines which has helped improve the quality of our work. We will include relevant benchmark results in the final version.

---

### Official Review · Reviewer_UeQu · 2024-07-13

**Soundness:** 3
**Presentation:** 2
**Contribution:** 3
**Rating:** 6
**Confidence:** 4

**Summary:**

The work proposes a new method that determines the factors of variation bonded to different labels in a supervised way, obtaining a method akin to supervised PCA but in several independent subspaces thorough the Hilbert-Schmidt independent criterion (that it’s employed to maximize both the independence between different subspaces and the dependence between the data subspaces and the labels). The paper focus on a linear kernel application of HSIC, but other kernels are also considered. Then it’s applied to some simulated data and to DNA/RNA sequencing data.

**Strengths:**

The theoretical connections are interesting.

The paper is technically sound, with a well-derived mathematical background.

The results of genetic sequence analysis are meaningful and interesting.

**Weaknesses:**

The novelty is limited, it appears to be an extension of previously described methods.

The comparison is provided only with PCA or sPCA, which are not by any means SOTA methods.

The explanation of the simulated data is not clear, what should it be the expected result? Are you learning the dimensions of the subspaces?

A better description of the experiments is needed. How do you choose the hyperparameters (for instance, the dimension of the subspaces is 10 in both sections 4.2 and 4.3, why?).

In section 4.2, the use of HSIC as a quality metric is not fair, since it’s already included in your loss.

The significance of the paper would be increased if the authors prove that the method is useful for the analysis of other types of real-world datasets.

The discussion is too short and not too clear. I acknowledge the limitations of space, but the discussion is one of the most important sections and should be properly done (even by moving some mathematical details to SI).

**Questions:**

Could you please provide a comparison of your method with other SOTA methods?

Would you prove that the method is useful in other kinds of data types?

Which are the hyperparameters of the method? How do you set them?

**Limitations:**

I’m not satisfied with the way the limitations are addressed. One should explain them clearly and provide examples in which the method is not working.

---

> ### Author Rebuttal · Authors · 2024-08-06
>
> ## **Responses to Reviewer UeQu**
> Thank you for your constructive feedback on the strengths and weaknesses of our work. We address your main concerns below:
>
> ## **Weaknesses**
> ### **W1: Limited novelty**
> While our method extends PCA, it offers significant contributions by uniquely combining supervision and disentanglement in a linear, interpretable framework valuable for feature selection. This addresses an important gap in current dimensionality reduction techniques, especially in an era where linearity and interpretability are often overlooked (**See M1**).
> ### **W2: Comparison with SOTA methods**
> We've added comparisons with non-linear VAE-based models (**See M2**), focusing on:
>
> 1. **Representation quality:** Low-dimensional representations capturing specific data aspects.
> 2. **Interpretability:** Understanding feature contributions to different data aspects.
>
> While the interpretability criterion clearly flavors linear models, our results show that linear models like sPCA are indeed strong baselines too in representation quality (**See M2.3, Fig S1, Table S2**). We also confirmed similar insights on supervision and disentanglement regularization effects across both linear and non-linear models.
>
> ### **W3: Inadequate discussion of limitations**
> We apologize for the poor presentation of limitations, which are somewhat scattered throughout the paper. We will concentrate them and improve the discussion section in the final version. Key limitations and their potential impacts include:
>
> 1. **The limitation of linear-kernel HSIC on theoretical independence guarantee** (Section 3.1, line 159 - 165). Though in practice this is less concerning, since minimizing HSIC-linear will also reduce HSIC-Gaussian (Section 4.2.3, Table 1).
> 2. **Subspace interpretability issue** (Section 3.3, line 189 – 193, and Appendix D on the resulting optimization landscape). This would make it especially challenging when the model needs to learn both supervised and unsupervised subspaces. Figure 7 is an example where unsupervised subspace failed to capture the true dimension.
> 3. **Challenges in balancing the disentanglement regularization strength \lambda** (Section 5). This is a general challenge to all supervised disentangled models.
> 4. **Potential performance loss due to the lack of non-linear feature interactions** (Section 3.1, line 166-169; Section 5). This could lead to underperformance on datasets with complicated patterns.
>
> ## **General questions**
> ### **Q1: How to set model hyperparameters, in particular the dimension of each subspace.**
> We again apologize for the lack of clarity. The main hyperparameters are subspace dimensions and disentanglement penalty scale λ (**See Table S1, M2.1**). The latter was discussed as a limitation in the Discussion and above in **W3**. For sPCA and sisPCA, subspace dimension generally doesn't affect performance beyond numerical issues from SVD solvers.
> - sPCA's subspace dimensions are determined by supervision kernel rank (Appendix B.3, line 406-408). That is, the effective dimension is always 1 for a 1-D continuous variable with linear kernel (age in Section 4.2) or K for a categorical variable of K groups with delta kernel (cancer type in Section 4.2) regardless of the specified dimension. The extra axes beyond the effective dimension (sPCA-age PC2-10) will collapse to zero since the eigenvalue is zero.
> - The effective dim of sisPCA subspaces is determined by both the supervision kernel rank as well as other subspaces (Remark 3.2, line 144-147). In practice, sisPCA shows similar behavior as sPCA, with extra dimensions typically collapsing to zero. For example, the sisPCA-infection subspace in Section 4.3 has approximately two effective dimensions (Figure 4, 9 and Figure S1).
> - In contrast, VAE models are more sensitive to dimension changes and have more hyperparameters to tune (**M2.1**), which is generally beyond the scope of our work. Due to time constraints, we mostly follow the SCVI’s default in designing VAE models for benchmark.
>
> ### **Q2: Better description of experiments**
> #### **Simulated data Figure 3**
> As mentioned in **Q1**, the dimensions of supervised subspaces are mostly predetermined by their target kernel ranks. Here dim(S1) and dim(S2) are 2 regardless of hyperparameter choice. The dimension of the unsupervised space S3 is unknown, and we aim to recover the ground truth donut structure (dim = 2) as did in Fig 3c. Figure 7 represents a failed example where only one dimension is recovered.
> #### **Subspace dimensions in Section 4.2 and 4.3**
> Addressed in **Q1**. Dimensions in PCA-based models are generally ranked by importance and can be selected using methods like "the elbow curve" (Discussion, line 311-313). In **M2** subspace dimensions of VAEs models are set to 10 to ensure fair comparison.
> #### **Applications to other types of real-world datasets**
> We note that similar to PCA and its extensions [1], our model is also general-purpose and can assist with data exploration tasks (*as pointed out by Reviewer debZ*). The only domain-specific model is the hsVAE-sc tailored for single-cell data with a count-based likelihood.
>
> [1] Abid et al., "Contrastive principal component analysis"
>
> ### **Q3: The use of HSIC as a quality metric in Section 4.2**
> The HSIC metrics in Table 1 are to support the claim that minimizing the HSIC-linear can also reduce the computationally more expensive HSIC-Gaussian (not explicitly optimized for in the loss) and thus encourage independence in the representations (line 259-261).

---

> > ### Comment · Reviewer_UeQu · 2024-08-11
> > **Follow up**
> >
> > I thank the authors for their reply and acknowledge their usefulness. I will, accordingly, raise my score.
> > However, I have to say that I'm not satisfied with the reply to "other kinds of data sets" reply and it refrains me from further raising the score.

---

> ### Author Response · Authors · 2024-08-12
> **New Application on Breast Cancer Diagnostic Data**
>
> We appreciate your feedback and apologize for our previous insufficient response. Due to space constraints, we couldn't include additional figures and results initially. To further illustrate sisPCA's versatility as a plug-in replacement of PCA for disentangling quantities, we provide below a new application, which will be included in the appendix of the final paper, and as a tutorial upon package release.
>
> ### **Problem and Dataset**
> We used the Kaggle Breast Cancer Wisconsin Data Set (569 samples, 30 summary features, uciml/breast-cancer-wisconsin-data, CC BY-NC-SA 4.0 license). The 30 real-valued features are computed from imaging data of breast mass, which include the mean, standard error and the extremum of quantities like cell nuclear radius, texture, perimeter etc.
>
> Here, our goals are to:
>
> 1.	Learn a representation for predicting sample disease status (Malignant or Benign, not used during training).
> 2.	Understand how original features contribute to data variability, the learned representation, and diagnosis potential.
>
> ### **Experiments and Results**
> We focus our comparison on three linear models: PCA, sPCA, and sisPCA, using zero-centered and variance-standardized data as inputs. The diagnosis label used to measure subspace quality remains invisible to all models. Below we summarize the quantitative results, and will include subspace visualization in the final paper.
>
> #### **PCA**
> We projected all features (dim = 30) onto one PCA subspace (dim = 6, determined by the elbow rule), explaining 61.6% of total variance. Malignant and benign samples appear well separated in PC1 and PC2. 'symmetry_mean' (loading = -0.223) and 'radius_mean' (loading = -0.219) are the top 2 features that negatively contribute to PC1. That is, the higher the two feature scores, the lower PC1 score and the greater the possibility of the sample being malignant.
>
> #### **sPCA**
> From the PC1 loadings, we sought to construct two subspaces to separately reflect nuclei size (‘radius_mean’ ) and shape ('symmetry_mean'). We set $Y_{radius}$ ('radius_mean', 'radius_sd') of $569 \times 2$ as the target variable for the radius subspace, and $Y_{symmetry}$ ('symmetry_mean', 'symmetry_sd') of $569 \times 2$ as the target for the symmetry subspace. The remaining 26 features were projected onto the two subspaces using sPCA (dim = 3, effective dim = 2). *Both subspaces better explained diagnosis status than PCA (**Table S3**) but remained highly entangled.* Specifically, the PC2 loadings of the two spaces have a Pearson correlation of 0.716, and 'perimeter_worst', which is highly correlated to 'radius_mean' (corr = 0.965), also strongly contributes to the PC2 of the symmetry subspace (loading = 0.238).
>
> #### **sisPCA**
> We next applied sisPCA ($\lambda$ = 10) to further disentangle the two subspaces, increasing their separation (Grassmann distance increased from 1.593 to 2.058, PC2 loadings correlation decreased to 0.257). Here ‘perimeter’ features no longer contribute to the symmetry subspace. *As a result, the radius subspace remained predictive of diagnosis, while the symmetry subspace became less relevant (**Table S3**).* We confirmed the finding by directly measuring the predictive potential of the target variables $Y_{radius}$ and $Y_{symmetry}$ (Silhouette score = 0.457 and 0.092, respectively).
>
> **Table S3: Predictability of diagnosis status, measured by Silhouette score**
>
> | | Radius subspace (dim = 3) | Symmetry subspace (dim = 3) | Overall subspace (dim = 6) |
> |:-|:-|:-|:-|
> | PCA (one subspace) | 0.294 (PC 1-3) | 0.013 (PC 4-6) | 0.160 (PC 1-6) |
> | sPCA | 0.534 | 0.374 | 0.464 |
> | sisPCA | 0.553 | 0.027 | 0.511 |
>
> ### **Interpretation**
> 1.	Our results suggest that nuclear size (radius subspace) is more informative for breast cancer diagnosis than nuclear shape (symmetry subspace), aligning with clinical observations [1].
> 2.	Unsupervised PCA and sPCA captured both radius and symmetry aspects as they contribute most to data variability. *Without disentanglement, the two aspects remain intertwined, potentially leading to incorrect conclusions about symmetry's predictive power for diagnosis.*
> 3.	*sisPCA successfully separated the two representations, revealing their distinct relationships to diagnosis.* The results are interpretable: the radius subspace is constructed using features like 'area' and 'perimeter', while the symmetry subspace uses features like 'compactness' and 'smoothness'.
>
> This example demonstrates sisPCA's ability to disentangle different aspects of data variation and uncover underlying relationships. **Importantly, sisPCA improves upon the diagnosis potential of PCA's representation, even through the diagnosis labels were never used during training.**
>
> [1] Kashyap, Anamika, et al. "Role of nuclear morphometry in breast cancer and its correlation with cytomorphological grading of breast cancer: A study of 64 cases." Journal of cytology 35.1 (2018): 41-45.

---

> > ### Comment · Reviewer_UeQu · 2024-08-13
> >
> > I thank the authors for their reply. This new application makes both the application and the method much clearer, and I would put it in the main text (just before 4.2). I will raise my score accordingly.

---

> > > ### Author Response · Authors · 2024-08-13
> > > **Thank you**
> > >
> > > Thank you for the feedback and suggestion. We will add this application as a new subsection and adjust the main text accordingly.

---

### Official Review · Reviewer_PWKM · 2024-07-21

**Soundness:** 3
**Presentation:** 3
**Contribution:** 3
**Rating:** 6
**Confidence:** 3

**Summary:**

This paper presents a method for distentangling multiple independent linear latent subspaces that align with a set of response variables. This uses the Hilbert-Schmidt Independence Criterion (HSIC) to measure the dependence each subspace and the targe variable providing supervision for that subspace, a concept which was previously applied to infer a single subspace that was connected to a set of response variables. HSIC is additionally used to enforce independence between the subspaces, to encourage the method to find independent subspaces that correlate with each of the provided dependent variables. This expands on Supervised PCA by enabling inference of an indendent subspace for each response variable, rather than a single subspace which aligns with the vector of response variables, reflecting the fact that different responses may be interacting with different subspaces in the data, e.g., methylation changes reflecting natural human aging and changes related to a disease like cancer.

The method is evaluated on synthetic data and shown to do a good job at extracting the different subspaces. In particular, Supervised PCA is dominated by one of the two supervised subspaces.

The method is then evaluated on real methylation data and single-cell gene expression data. In the cancer data, the new method is shown to provide subspaces with greater separatio

**Strengths:**

The paper tackles an important problem in the analysis of high-dimensional omics data, where there is a need to identify interpretable subspaces that can be mapped onto higher-level biological processes but where different phenotypes (e.g., aging vs cancer) may manifest differently, requiring independent subspaces rather than supervision of a single subspace as per sPCA.

The paper is well-written and combines simulated data experiments, which help understand the method is behaving as expected, with two applications on real data.

**Weaknesses:**

One could argue that the work is somewhat incremental. The use of the HSIC is not new and nor is the idea of providing supervision to a latent variable model to ensure the latent subspaces reflected some additional variables. However, I feel the contribution is novel enough to warrant presentation at NeurIPS

**Questions:**

No questions

**Limitations:**

The authors describe the well-known limitations of unsupervised PCA, i.e., identifiability, and other limitations implied by their choice of kernel.

---

> ### Author Rebuttal · Authors · 2024-08-06
>
> ## **Response to Reviewer PWKM**
> Thank you for your summary. We appreciate your evaluation that our technical contribution is novel enough to warrant presentation at NeurIPS.

---

### Author Rebuttal · Authors · 2024-08-06

We thank the reviewers for their thoughtful comments. We first address the two main points raised by multiple reviewers, followed by responses to individual comments.

### **M1: Highlight the technical novelty (Reviewers PWKM and UeQu)**
We appreciate your acknowledgment of our paper’s technical contribution as novel and sensible (Reviewer debZ). To further clarify our design insights:
- **Linearity and interpretability by design.** PCA remains popular due to its interpretability. It is particularly valuable in data exploration, where the goal is to identify features that best represent the data in specific aspects. In this regard, linear models are indeed the state-of-the-art (See **M2.2**). We motivated our work as a multi-subspace extension of PCA to fully leverage its interpretability. It is the first linear supervised disentanglement model and is competitive with or even outperform non-linear counterparts (See **M2.3**).
- **Theoretical advantages from linearity.** We show that sisPCA can be viewed both as linear auto-encoder (Remark 3.1 and Appendix B.1) and as regression for continuous targets (Remark 3.3 and Appendix B.3). Linearity also allows for more efficient and reliable optimization (Algorithm 1 and Conjecture 3.1).

### **M2: Comparison with non-linear SOTA methods (Reviewers UeQu and QSK9)**
We have now included additional baselines based on the HSIC-constrained VAE (HCV) ([1] of Reviewer QSK9).

### **M2.1 SOTA implementation**
We reimplemented the idea from [1] using the latest SCVI framework (scvi-tools v1.1.2) for variational inference training and designed three non-linear counterparts of PCA models:
1. **VAE**: Vanilla VAE with Gaussian likelihood.
2. **sVAE**: VAE with additional predictors for target variables.
3. **hsVAE**: sVAE with additional HSIC penalty.

Non-linear models generally have more hyperparameters. In our benchmark, we fixed the VAE architecture following the SCVI default (e.g. one layer of NN with 128 hidden units and batch normalization), or the scVIGenQCModel in [1] (e.g. equal weighting of prediction and reconstruction losses).

**Table S1: General model comparison**
| | Linear ||| Non-linear |||
|:--|:--:|:--:|:--:|:--:|:--:|:--:|
| | PCA | sPCA | sisPCA (this work) | VAE | sVAE | hsVAE |
| Supervision | - | HSIC | HSIC | - | NN prediction | NN prediction|
| Disentanglement | - | - | HSIC | - | - | HSIC |
| Interpretation | Linear projection *U* as feature importance ||| Blackbox ||
| Hyperparameters | 1) #dim | 1) Subspace #dim. 2) Target kernel choice | 1) and 2) of sPCA. 3) HSIC penalty | 1) #dim. 2) General NN design. | 1) and 2) of VAE. 3) Predictor design | 1), 2) and 3) of sVAE. 4) HSIC penalty|
| Optimization | Closed form | | Simple landscape | General limitations of NN and VI like robustness** ||

\** We were not able to run VAEs on the 6-dimensional simulated data in Figure 3 due to NaNs generated during variational training, which is not uncommon for SCVI models and is likely the result of exploding parameters.


### **M2.2 Interpretability**
We define and compare model interpretability based on the practical need to *learn how each feature contributes to specific aspects of the high-dimensional data*.
Linear models, as demonstrated in Section 4.3.3, are inherently interpretable because the learned projection *U* directly shows how each subspace axis relates to original features. PCA-based models have the additional advantage of ordering subspace axes based on importance (variance explained). In Figure 5, we selected subspace-specific genes based on PC1 loading scores, which would not be possible without the mapping *U*.
In contrast, VAE models link features and subspace axes through a non-linear black box. While some interpretation approaches such as gradient-based saliency map are becoming standard practice, they are nowhere near as straightforward as the linear mapping. In addition, since gradient and loss can only be calculated per sample, aggregating them into a global feature importance score is non-trivial.

### **M2.3 Results on the single-cell infection data**
We have now included results from VAE-based models (**see attached Fig S1 for visualization**). We also provide below quantitative metrics in **Table S2**. While higher scores generally indicate better results, we caution that these metrics may not fully capture the biological relevance of the representations. As in Section 4.3.2, our analysis aims to isolate genes directly involved in parasite harboring from those associated with broader temporal changes post-infection. A better temporal subspace should tell us about subtle progression (pseudo-time) even of the same collection time beyond prediction accuracy (since we already knew the labels).

**Table S2: Quantitative evaluation of representation quality**

| Method | Linear ||| Non-linear ||||
|:--|:--:|:--:|:--:|:--:|:--:|:--:|:--:|
| | PCA | sPCA | sisPCA (this work) | VAE | sVAE | hsVAE | hsVAE-sc |
| Subspace separateness-Grassmann | 0 | 3.802 | 4.467 | 0 | 3.510 | 3.797 | 3.459 |
| Information density-Silhouette: (infection, time) | (0.068, 0.083) | (0.313, 0.279) | (0.294, 0.164) | (0.045, 0.137) | (0.058, 0.215) | (0.064, 0.197) | (0.235, 0.353) |
| Information density after UMAP-Silhouette: (infection, time) | (0.009, 0.238) | (0.153, 0.364) | (0.197, 0.228) | (-0.022, 0.335) | (0.008, 0.602) | (0.047, 0.586) | (0.258, 0.603) |

Overall, we've confirmed that:

- HSIC against target efficiently imposes supervision, achieving performance equivalent to neural networks and making sPCA indeed a strong baseline.
- sisPCA outperforms its general-purpose VAE counterparts, particularly in the infection subspace. However, VAEs can gain further SOTA performance from domain-specific knowledge (such as count-based modeling).
- Disentanglement may come at the cost of weakening supervision, aligning with Section 4.2.3 and Table 2.

We will include the above results (**Fig S1, Table S1 and S2**) in the final version.

---

### Decision · Program_Chairs · 2024-09-25

**Decision:**

Accept (poster)

**Comment:**

All four reviewers argued to accept the paper, albeit three of them only weakly. However, the paper has improved quite a bit relative to initial reviewer concerns during the discussion period, particularly noting the additional experiment which broadens the scope from the focus on single-cell RNA sequencing data; and the study into automatic settings of \lambda and the sensitivity to the parameter choice. Multiple reviewers raised their score after the author response.

Please be certain to incorporate everything discussed in the rebuttal phase when preparing the final version of the manuscript.